# Caspase 3 and caspase 7 promote cytoprotective autophagy and the DNA damage response during non-lethal stress conditions in human breast cancer cells

**Gayathri Samarasekera**[1,2], Nancy E. Go[1], Courtney Choutka[1,3], Jing Xu[1], Yuka Takemon[1,4,5], Jennifer Chan[1,3], Michelle Chan[1,3], Shivani Perera[1], Samuel Aparicio[6,7], Gregg B. Morin[1,2], Marco A. Marra[1,2,5], Suganthi Chittaranjan[1], Sharon M. Gorski[1,2,3,8]*

1 Canada's Michael Smith Genome Sciences Centre, BC Cancer, Vancouver, British Columbia, Canada, 2 Department of Medical Genetics, University of British Columbia, Vancouver, British Columbia, Canada, 3 Department of Molecular Biology and Biochemistry, Simon Fraser University, Burnaby, British Columbia, Canada, 4 Genome Science and Technology Graduate Program, University of British Columbia, Vancouver, British Columbia, Canada, 5 Michael Smith Laboratories, University of British Columbia, Vancouver, British Columbia, Canada, 6 Department of Molecular Oncology, BC Cancer, Vancouver, British Columbia, Canada, 7 Department of Pathology and Laboratory Medicine, University of British Columbia, Vancouver, British Columbia, Canada, 8 Centre for Cell Biology, Development and Disease, Simon Fraser University, Burnaby, British Columbia, Canada

* sgorski@bcgsc.ca

## Abstract

Cell stress adaptation plays a key role in normal development and in various diseases including cancer. Caspases are activated in response to cell stress, and growing evidence supports their function in non-apoptotic cellular processes. A role for effector caspases in promoting stress-induced cytoprotective autophagy was demonstrated in *Drosophila*, but has not been explored in the context of human cells. We found a functionally conserved role for effector caspase 3 (CASP3) and caspase 7 (CASP7) in promoting starvation or proteasome inhibition-induced cytoprotective autophagy in human breast cancer cells. The loss of CASP3 and CASP7 resulted in an increase in PARP1 cleavage, reduction in LC3B and ATG7 transcript levels, and a reduction in H2AX phosphorylation, consistent with a block in autophagy and DNA damage-induced stress response pathways. Surprisingly, in non-lethal cell stress conditions, CASP7 underwent non-canonical processing at two calpain cleavage sites flanking a PARP1 exosite, resulting in stable CASP7-p29/p30 fragments. Expression of CASP7-p29/p30 fragment(s) could rescue H2AX phosphorylation in the CASP3 and CASP7 double knockout background. Strikingly, yet consistent with these phenotypes, the loss of CASP3 and CASP7 exhibited synthetic lethality with BRCA1 loss. These findings support a role for human caspases in stress adaptation through PARP1 modulation and reveal new therapeutic avenues for investigation.

## Introduction

Caspases are cysteine-dependent aspartic proteases and are traditionally known for their role in proteolysis during the final stages of apoptosis. It is increasingly appreciated that caspases

**Data availability statement:** All relevant data are within the paper and its Supporting information files. The code is publicly available in a GitHub repository (https://github.com/MarraLab/Caspase_GRETTA_analysis) and archived on Zenodo (https://doi.org/10.5281/zenodo.14722298).

**Funding:** CC was supported by a NSERC Doctoral Postgraduate Scholarship and JX was supported by a CIHR Frederick Banting and Charles Best Canada Graduate Scholarship Doctoral Award. This research was supported by a Canadian Institutes of Health Research (CIHR) Operating grant (MOP-78882), CIHR in partnership with Avon Foundation for Women-Canada grant (OBC127216), CIHR Project grants (PJT-159536, PJT-191846), and CIHR bridge grants (PLL-185685, PLL-190346) all to SMG and GBM. The funders did not play any role in the study design, data collection and analysis, decision to publish, or preparation of the manuscript.

**Competing interests:** I have read the journal's policy and the authors of this manuscript have the following competing interests: GS, NEG, CC, YT, MAM, SC, and SMG are co-inventors, but have no right, title or interest, on a US Provisional Patent Application 63/587254 entitled Compositions and methods for inhibition of CASP3 and CASP7, and on a US Patent Application CA2024051308 entitled Compositions and methods for inhibition of CASP3 and CASP7 for cancer therapy. SA is a founder of Inflex Ltd., outside the scope of this study. The remaining authors have declared that no competing interests exist.

**Abbreviations:** BafA1, Bafilomycin A1; CASP3, caspase 3; CASP7, caspase 7; CASP8, caspase 8; DDR, DNA damage response; DepMap, Dependency Map; DKD, double knockdown; DKO, double knockout; DMEM, Dulbecco's Modified Eagle Medium; EBSS, Earle's Balanced Salt Solution; FBS, fetal bovine serum; GRETTA, Genetic inteRaction and EssenTiality neTwork mApper; HR, homologous recombination; IP, immunoprecipitation; MOMP, mitochondrial outer membrane permeabilization; PARP1, poly(ADP ribose) polymerase 1; PBS, phosphate-buffered saline; PDX, patient-derived xenograft; PI, proteasome inhibitor; PVDF, polyvinylidene difluoride;  RT, room temperature; RT-qPCR, reverse transcription-quantitative polymerase chain reaction; KD, single knockdown; KO, single knockout.

also play important roles in non-apoptotic cellular processes in normal development and in disease conditions [1–4]. Given the evolutionarily conserved non-apoptotic caspase roles across phyla, it has been postulated that the primordial function of caspases was to regulate cellular stress adaptations [1,5,6]. Further supporting this notion are the distinct caspase 3 (CASP3)- and caspase 7 (CASP7)-dependent proteolytic landscapes in cells exposed to non-lethal stress conditions compared to lethal stress conditions [7]. Such adaptive roles under non-apoptotic stress may explain unexpected observations, such as the association of high caspase expression with enhanced tumor progression or the lack of correlation between caspase expression and apoptosis in several cancer types, including breast cancer [8–15]. Caspases are ubiquitously expressed in most cells and several studies have linked caspases to stress response pathways [16–19]. Interestingly, stimuli that activate cell stress response pathways can also activate caspases [20,21]. However, the processing, regulation, and activity of caspases during non-lethal cellular stress are understudied.

Macroautophagy, hereafter referred to as autophagy, is an evolutionarily conserved intra-cellular lysosome-mediated degradation and recycling process that plays significant roles in normal development, aging and diseases, including cancer [22–24]. While autophagy occurs at basal levels, it is also a major pathway upregulated in response to several stressors, including nutrient deprivation, hypoxia, reactive oxygen species, DNA damage, and pathogens [25–27]. Autophagy supports stress adaptation and cell survival through degrading and recycling damaged organelles and macromolecules to facilitate the production of energy and/or essential cellular components [27,28].

The mechanistic interactions between cytoprotective autophagy and apoptosis are not well-understood. It is widely accepted that these two processes are antagonistic, and hence the final cell fate is determined by a tug-of-war between pathways [29,30]. Consistent with this model, caspases were shown to suppress autophagy by direct cleavage of autophagy regulators or core autophagy proteins [29,31]. However, apoptotic pathway components, including caspases, have also been implicated in promoting autophagy in some contexts [31–33]. In *Drosophila*, the apoptotic effector caspase Dcp-1 was shown to positively regulate stress-induced cytoprotective autophagy [34–36]. In human cells, a role in the positive regulation of stress-induced cytoprotective autophagy was demonstrated for the initiator caspase 9 [32]. However, it is unknown whether human effector caspases have an evolutionarily conserved function in stress-induced cytoprotective autophagy. In addition, the contexts, molecular mechanisms and pathways involved in non-lethal stress adaptation by both initiator and effector caspases have not been thoroughly investigated.

All caspases exist as inactive zymogens (pro-caspases) consisting of a pro-domain, a large ~ p20 subunit containing the catalytic site, a small ~ p10/p12 subunit, and may contain a linker sequence between subunits. Upon activation in apoptosis, the pro-domain and the linker region are removed by proteolytic cleavage in a temporal order unique to each caspase, ultimately resulting in ~ p20 and p10/p12 cleaved caspase fragments that assemble to form the active tetrameric complex [37]. In effector caspases, these cleavage events are typically carried out by initiator caspases. Granzyme B and calpain have also been implicated [38–40]. Once activated, effector caspases cleave many substrate proteins [41]. Therefore, caspase activity must be tightly regulated, presumably in both apoptotic and non-apoptotic settings. One major outstanding question is how caspases participate in non-apoptotic functions without killing cells. Identified modes of caspase regulation include post-translational modifications (cleavage, phosphorylation, ubiquitylation), subcellular compartmentalization (mitochondria, cytosol, nucleus, stress granules), and modulatory protein interactions, including inhibitor of apoptosis protein family members [5,6,19,42]. Allosteric sites in CASP3 and CASP7 and an exosite (non-active site interaction) in CASP7 can also contribute to regulation and activity

[43]. However, caspase processing, activation, and regulation in non-apoptotic functions and/ or in non-lethal cellular stress conditions have not been well characterized.

Here, we demonstrate that human effector CASP3 and CASP7 have an evolutionary conserved role in promoting cytoprotective autophagy during non-lethal stress. We found that the underlying mechanism involves non-canonical cleavage of CASP7 and alteration of Poly(ADP ribose) polymerase 1 (PARP1) activity. The combined loss of CASP3 and CASP7 phenocopies PARP1-inhibition, including synthetic lethality with BRCA1 loss.

## Results

### Dual loss of CASP3 and CASP7 suppresses starvation-induced autophagy

We used the autophagy-activating role in response to nutrient deprivation of the *Drosophila* effector caspase Dcp-1 [34–36] as the basis for investigating the potential functional conservation between two closely related human caspases, effector CASP3 and CASP7. First, we employed standard MAP1LC3B/LC3B-based autophagy assays as these have been used successfully to monitor autophagy in breast cancer cell lines, including in SKBR3 lines, and under similar stress conditions we employed [44–47]. To determine the optimum time point for detecting a consistent upregulation of autophagy without signs of cell death, SKBR3 cells were subjected to amino acid starvation (Fig 1A and 1B). We observed a significant increase in autophagic flux at 8- and 24-h starvation, as measured by the increased LC3BII accumulation in the presence of BafA1 (Fig 1B). Across all time points, low levels of cleaved-PARP (at ~89 kDa) were observed in both fed (non-stressed) control cells and starved (stressed) cells as reported previously as being normal even in non-apoptotic viable cells [7,48]. Furthermore, there were no signs of cell death, as indicated by the absence of a marked increase in cleaved-PARP1 in the starvation conditions (Fig 1A), and thus we selected 8-h starvation as the optimal time point for detecting non-lethal stress-related starvation-induced autophagic flux in SKBR3 cells in this study. Next, to determine whether amino acid starvation-induced autophagy was CASP3- and/or CASP7-dependent, single knockdown (KD) and double knockdown (DKD) experiments were performed with siRNAs. The control scramble-siRNA-treated cells showed an increase in LC3BII levels in the presence of BafA1, indicating an increase in autophagic flux when cells were starved (Fig 1C). In single CASP3 or CASP7 siRNA-treated cells, the starvation-induced autophagic flux was comparable to that of the control. However, a significant suppression of autophagic flux was observed in CASP3 and CASP7 (CASP3 + 7) DKD cells (Fig 1C and 1D).

We hypothesized that the lack of a significant effect on starvation-induced autophagy upregulation in single CASP3 or CASP7 KD experiments might be due to the presence of residual caspase levels (S1A and S1B Fig). To address this, we generated CASP3 and CASP7 single knockout (KO) and double knockout (DKO) SKBR3 cell lines (S1C Fig). In single caspase KO cells, we found negligible or undetectable effects on starvation-induced autophagy compared to the control parental cell lines (Fig 1E and 1F). This suggests that CASP3 and CASP7 have partially overlapping functions and/or single KO cell lines may have activated compensatory mechanisms to overcome the other caspase's loss. Consistent with this notion and with what we observed in DKD cells, there was a significant impairment of autophagic flux in CASP3 + 7 DKO cell lines (Fig 1E and 1F). Similarly, following 24 hours starvation, there was no induction of autophagic flux in CASP3 + 7 DKO cells (S1D and S1E Fig). To confirm the specificity of our findings, we generated multiple isogenic caspase DKO SKBR3 cell lines. In all five CASP3 + 7 DKO lines we tested, we observed a reduction in LC3B-II levels (in the presence of BafA1) indicating that amino acid starvation-induced autophagy was compromised (S1F Fig). To exclude the possibility of cell line-specific effects, we repeated our experiments using another

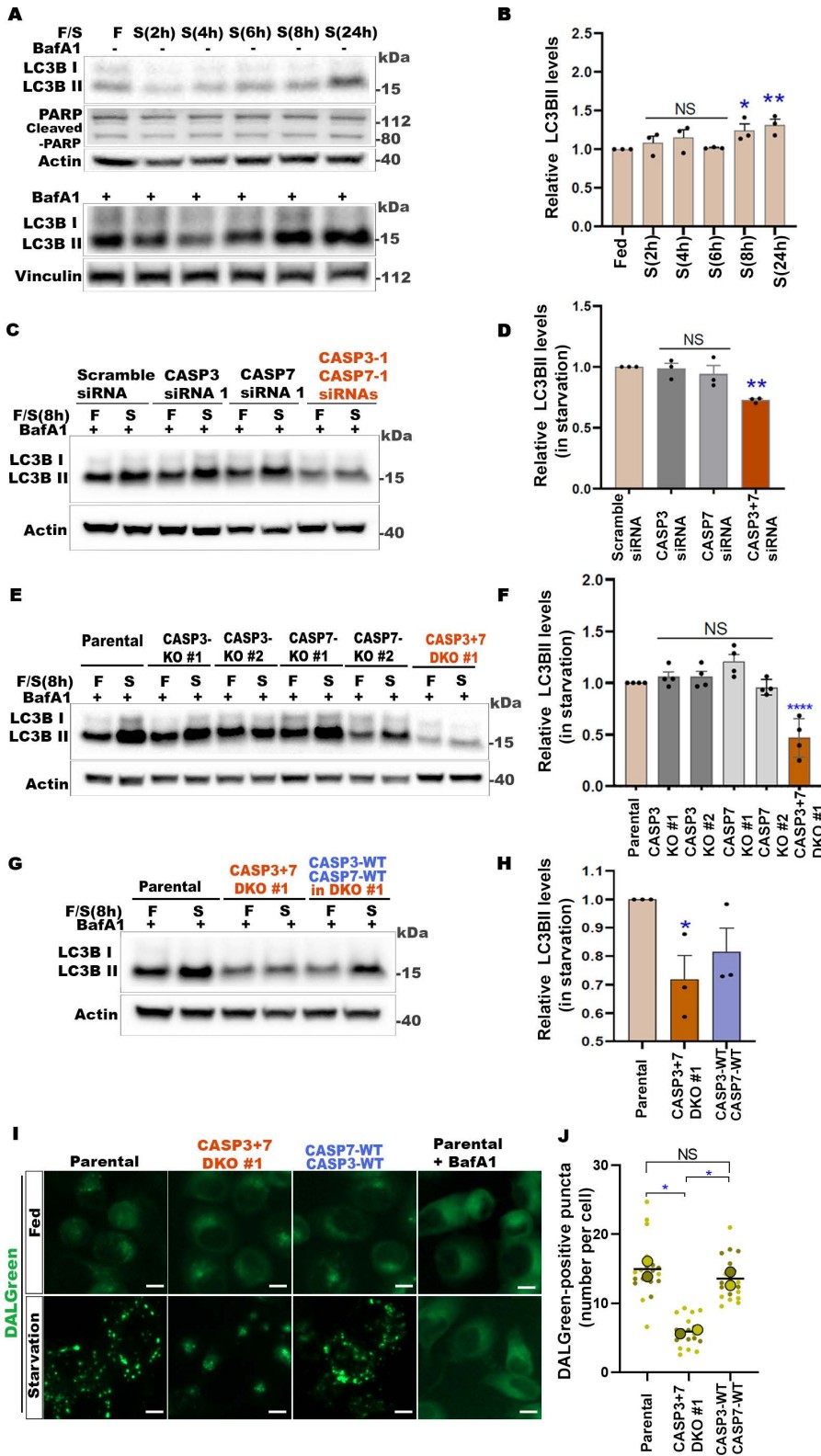

**Fig 1. Dual loss of CASP3 and CASP7 suppresses starvation-induced autophagy.** (**A**) Representative western blots of indicated proteins from SKBR3 cells in fed (F) or in amino acid starvation (S) for various time periods, in the absence or presence of 50 nM Bafilomycin A1 (BafA1) in the final 2 h. (**B**) Quantification of LC3B-based autophagy

flux in starved cells relative to the fed control, shown in (A). All with BafA1. (**C**) Representative western blots of indicated proteins from SKBR3 cells transfected with scramble, CASP3 and/or CASP7 siRNAs (48 h) and incubated in fed conditions or starved for 8 h with BafA1 (50 nM) for the final 2 h. (**D**) Quantification of LC3BII-based autophagy flux in starved cells relative to the starved scramble-siRNA control, shown in (C). (**E**) Representative western blots of indicated proteins from CASP3, CASP7 single (KO), or double knockout (DKO) SKBR3 cells in fed or starved (8 h) conditions, with BafA1 (50 nM) in the final 2 h. (**F**) Quantification of LC3BII-based autophagy flux in starved cells relative to the parental control, shown in (E). (**G**) Representative western blots of indicated proteins from SKBR3 parental, DKO or CASP3 + 7-WT re-expression in DKO cells in fed or starved (8 h) conditions, with BafA1 (50 nM) in the final 2 h. (**H**) Quantification of LC3BII-based autophagy flux in starved cells relative to starved parental cells, shown in (G). (**I**) Representative immunofluorescence images of SKBR3 parental, DKO or CASP3 + 7-WT re-expression in DKO cells treated with 0.25 μM DALGreen in fed or starved (8 h) conditions. BafA1 (50 nM for 8 h) in both fed and starved conditions serve as controls. Scale bars, 20 μm. (**J**) Quantification of DALGreen immunofluorescence in starved cells shown in (I). Graph shows number of punctae per cell. $n = 2$, each with 10 random images covering total of 500–700 cells. In graphs, data are shown as mean ± SEM. $n =$ at least 3 independent experiments except in (J). *$P < 0.05$, **$P < 0.01$, ***$P < 0.001$, ****$P < 0.0001$, NS, not significant. In B, D, F, and H, one-way ANOVA with Dunnett's post-test. In J, one-way ANOVA with Tukey's post-test. See also S1 Fig. The numerical data presented in this figure can be found in S1 Data.

breast cancer cell line, MDA-MB-231, and similarly found that autophagic flux induction was significantly compromised only upon CASP3 + 7 DKD and DKO (S1G–S1L Fig). Lastly, when wild-type constructs of both CASP3 and CASP7 were stably reintroduced into DKO cells, the starvation-induced upregulation of autophagy was partially rescued (Fig 1G and 1H).

The processing of LC3B to form LC3BII is a hallmark of autophagy [49]. However, since LC3B is also involved in autophagy-independent processes [50], we orthogonally measured autophagy by employing the DALGreen autolysosome fluorescent marker [51]. In accordance with the LC3B-based autophagy assay results, the levels of DALGreen positive puncta indicate that amino-acid deprivation-induced autophagy is significantly compromised in CASP3 + 7 DKO cells, and the dual re-expression of CASP3 and CASP7 fully rescued autophagy (Fig 1I and 1J). The reduction in number of autolysosomes in CASP3 + 7 DKO cells was not due to a difference in cell size (S1M and S1N Fig). These observations together indicate that CASP3 and CASP7 play a partially redundant positive regulatory role in non-lethal amino acid deprivation-induced autophagy.

## Dual loss of CASP3 and CASP7 suppresses proteasome inhibition-induced compensatory autophagy and sensitizes cells to proteasome inhibitors

It is well-documented that autophagy is upregulated as a compensatory mechanism to maintain protein homeostasis, when proteasome function is compromised [52,53]. Next, we investigated whether this compensatory autophagy was also dependent on CASP3 and CASP7. A proteasome inhibitor (PI) MG132, at 0.5 μM, was chosen as the non-lethal stress condition for these experiments. (Figs 2A–2C and S2A–S2C). The reduction of proteasome activity was confirmed directly by proteasome activity assays (Figs 2C and S2C) and indirectly through the accumulation of ubiquitinated proteins (Figs 2A and S2A). In line with what we observed in amino acid starvation conditions, the PI-induced autophagic flux was significantly inhibited in CASP3 + 7 DKD or DKO cells (Fig 2D–2G). Although the effect was modest, single CASP7 KO also resulted in a significant reduction in PI-induced autophagy (Fig 2F and 2G). Similar results were observed in the MDA-MB-231 cell line (S2D and S2E Fig), and with the clinically approved proteasome inhibitor Bortezomib [54,55] (S2F and S2G Fig). PI-induced LC3BII upregulation was partially rescued by dual re-expression of CASP3 + 7, supporting the requirement for these effector caspases in PI-induced autophagy (Fig 2H and 2I). Collectively,

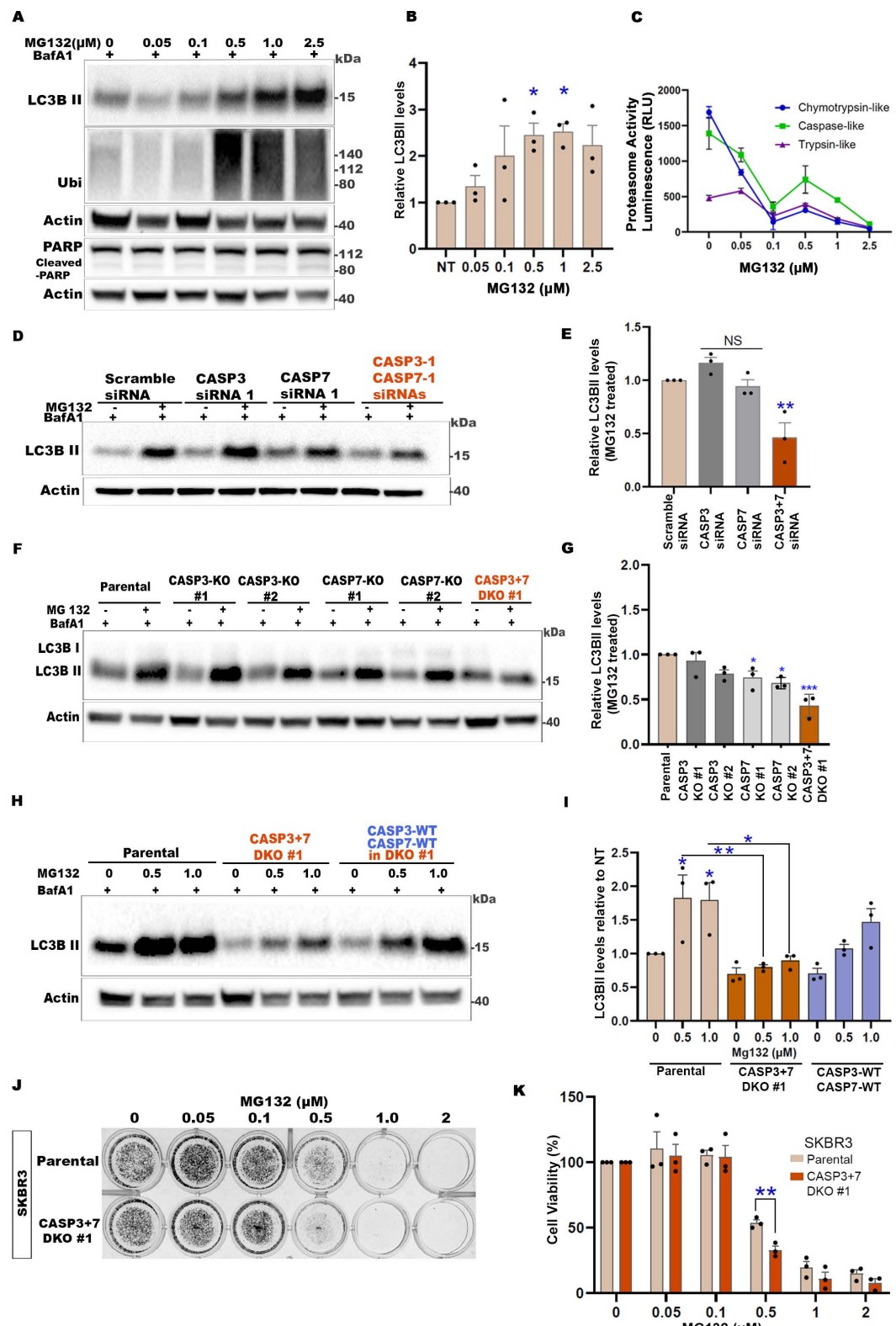

**Fig 2. Dual loss of CASP3 and CASP7 suppresses proteasome inhibition-induced compensatory autophagy and sensitizes cells to proteasome inhibitors.** (**A**) Representative western blots of indicated proteins from SKBR3 cells treated with MG132 at increasing dosage for 24 h, with BafA1 (50 nM) in the final 2 h. (**B**) Quantification of LC3B-based autophagy

flux in MG132-treated cells relative to untreated (0 µM MG132; DMSO vehicle only; NT) SKBR3 shown in (A). The levels of LC3BII were normalized to loading control and shown relative to the untreated control (NT). (**C**) Graph showing proteasome activity in response to increasing concentrations of MG132 as depicted by chymotrypsin, caspase and trypsin-like activities. (**D**) Representative western blots of indicated proteins from SKBR3 cells transfected with scramble, CASP3 and/or CASP7 siRNAs (48 h) and treated with vehicle DMSO only or MG132 (0.5 µM) for 24 h, with BafA1 (50 nM) in the final 2 h. (**E**) Quantification of LC3B-based autophagy flux in MG132-treated cells relative to the MG132-treated scramble-siRNA control, shown in (D). (**F**) Representative western blots of indicated proteins from CASP3, CASP7 single knockout, or DKO SKBR3 cells treated with vehicle DMSO or MG132 (0.5 µM) for 24 h, with BafA1 (50 nM) in the final 2 h. (**G**) Quantification of LC3B-based autophagy flux in MG132-treated cells relative to the MG132-treated parental control show in (F). (**H**) Representative western blots of indicated proteins from SKBR3 parental, DKO or CASP3 + 7-WT reintroduced into DKO cells treated with vehicle DMSO or MG132 (0.5 or 1.0 µM) for 24 h with BafA1 (50 nM) in the final 2 h. (**I**) Quantification of LC3BII-based autophagy flux in MG132-treated cells relative to DMSO-treated parental control, shown in (H). (**J**) Representative images of crystal violet assay plates of SKBR3 parental and CASP3 + 7 DKO cells treated with indicated concentrations of MG132 for 24 h and continued to grow in drug free media for another 3 days. (**K**) Quantification of cell viability shown in (J). Percentage of stained (viable) cells at each concentration was normalized to respective untreated cells. In graphs, data are shown as mean ± SEM. $n$ = 3 independent experiments. * $P < 0.05$, ** $P < 0.01$, *** $P < 0.001$, **** $P < 0.0001$, NS, not significant. In B, C, E, and G, one-way ANOVA with Dunnett's post-test and I with Tukey's post-test. K with two-way ANOVA with Sidak's post-test. See also S2 Fig. The numerical data presented in this figure can be found in S1 Data.

these results indicate that CASP3 and CASP7 have positive regulatory roles in autophagy induction in response to starvation or proteasome inhibition-induced stress.

Since autophagy has been reported to promote cell death in some contexts [56–58], we investigated whether the CASP3- and CASP7-dependent stress-induced autophagy contributes to cell death or survival. Crystal violet cell viability assays were carried out following PI treatment (MG132 or Bortezomib) or starvation of parental and CASP3 + 7 DKO or DKD of SKBR3 or MDA-MB-231 cells (Figs 2J, 2K and S2H–S2O). We observed a modest, but significant reduction in cell viability in response to PI treatment or starvation in CASP3 + 7 DKD or DKO cells compared to control cells. Contrary to traditional caspase roles, these results suggest that CASP3 and 7 promote cell survival in non-lethal PI or starvation stress conditions.

## CASP7 is cleaved non-canonically generating stable p29/p30 fragments under non-lethal stress conditions

To elucidate the molecular mechanisms by which CASP3 and CASP7 positively regulate cytoprotective autophagy, we analyzed their expression and processing pattern following amino acid starvation or PI treatments. CASP3 and CASP7 processing during apoptosis each result in large ~ p20 and small ~ p10/p12 cleaved-caspase fragments [37,59,60]. Intriguingly, our western blot analyses revealed a stable non-canonical CASP7 cleavage fragment(s) at ~ 30 kDa (p30) and/or 29 kDa (p29), and a CASP3 cleavage fragment at ~ 27 kDa (p27) in non-lethal starvation (Fig 3A and 3B) or PI conditions (Fig 3C and 3D). In fed or untreated (non-stressed) cells, these non-canonical fragments were either absent or less apparent. The CASP3 fragment at ~ 27 kDa was inconsistent and/or not readily detectable (e.g., Fig 3C) across all experiments, so we focused on the CASP7-p29/p30 fragments.

To confirm and further explore the observed non-canonical processing, we analyzed CASP7 processing in different contexts. We compared the CASP7 processing profiles resulting from starvation stress to apoptosis-induced profiles (staurosporine-treated cells) (Fig 3E). A marked increase in cleaved-PARP and canonical cleaved-CASP3 (p20) were present only in staurosporine-treated cells, confirming that the amino acid starvation conditions employed were non-lethal. The CASP7-p29/30 fragment(s) were produced predominantly only in the

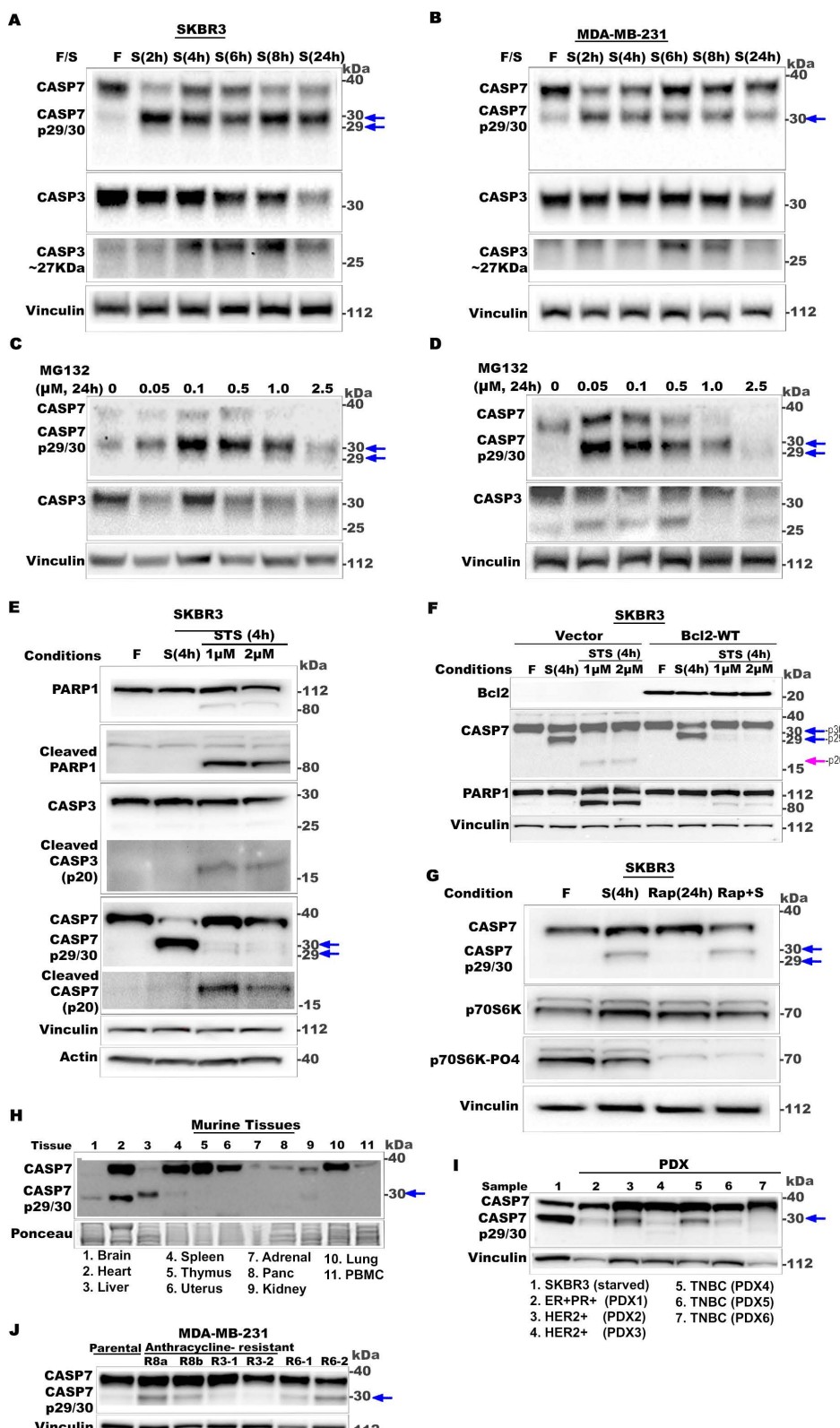

**Fig 3. CASP7 is cleaved non-canonically generating stable p29/p30 fragments under non-lethal stress conditions. (A–D)** Representative western blots showing changes in CASP7 and CASP3 in SKBR3 or MDA-MB-231 cells in amino acid starvation (A, B) or treated with MG132 (C, D). Non-canonical CASP7 bands (p29/p30) at 30 kDa

are indicated by blue arrows. A non-canonical CASP3 band (p27) at 27 kDa is also detected in some conditions. (**E**) Representative western blots comparing CASP7, CASP3, and PARP1 processing patterns in SKBR3 cells following 4-h incubation in fed conditions (F), non-lethal stress (starvation) or lethal stress (apoptosis induction by staurosporine; STS, 1 or 2 μM). (**F**) Representative western blots comparing CASP7 and PARP1 processing in SKBR3 parental cells stably transfected with vector control or BCL2 expression construct, following 4-h incubation in fed conditions (F), non-lethal stress (starvation) or lethal stress (apoptosis induction by staurosporine; STS, 1 or 2 μM). (**G**) Representative western blots showing the effect of rapamycin on the formation of CASP7-p29/p30 bands in SKBR3 cells. Cells in fed, starved (4 h), and/or treated with 10 nM rapamycin (Rap; 24 h). Blot was immunolabeled with mTOR activity reporters P70S60K and p70S6K-PO4. (**H**) Representative western blot showing CASP7 immunolabeling in tissues from a female CD-1 mouse. Ponceau S labeling serves as the loading control. *n* = 3 mice. (**I**) Representative western blot showing CASP7 immunolabeling in human breast cancer PDX specimens from indicated subtypes. Starved SKBR3 serves as the positive control for p29/p30 fragments. (**J**) Representative western blot showing CASP7 immunolabeling in parental MDA-MB-231 cells and derivative epirubicin-resistant (R8a, R8b, R3-1, R3-2) and 5Fu-resistant (R6-1, R6-2) cells in fed conditions. In each, at least *n* = 2 independent experiments. See also S3 Fig.

non-lethal starvation conditions, whereas the canonical apoptosis-associated CASP7-p20 fragments were primarily observed in staurosporine-treated cells. Altogether, these results confirm that the CASP7 processing in response to non-lethal stress conditions is distinct from CASP7 processing during apoptosis.

Mitochondrial outer membrane permeabilization (MOMP) activates caspases in apoptotic and non-apoptotic conditions [61,62]. BCL2 overexpression inhibits MOMP and subsequent CASP3 and CASP 7 processing in apoptosis [63,64]. To investigate whether MOMP plays a role in CASP7 non-canonical processing in non-lethal stress, we overexpressed BCL2 in SKBR3 parental cells. As expected, BCL2 overexpression inhibited canonical CASP7 processing in the control staurosporine-treated cells. In contrast, BCL2 overexpression did not inhibit the formation of CASP7-p29/30 in the non-lethal starvation stress condition (Fig 3F). These results indicate that CASP7 non-canonical processing is independent of MOMP.

We next asked whether other known autophagy inducers would also generate non-canonical CASP7 cleavage. Rapamycin inhibits mTOR and induces autophagy [65,66]. In SKBR3 and MDA-MB-231 cell lines (S3A and S3B Fig), autophagy was induced upon treatment with rapamycin. However, rapamycin did not result in non-canonical CASP7 processing (Figs 3G and S3C). Further, CASP3 + 7 DKO had no effect on rapamycin-induced autophagy (S3A and S3B Fig). Collectively, these observations suggest that caspases act upstream or in parallel to mTOR in autophagy regulation.

Next, we investigated the prevalence and the context dependency of non-canonical CASP7 processing. Using fed (unstressed) mice, we confirmed the existence of in vivo non-canonical processing of CASP7 in several tissues, including in the brain, heart, spleen, and kidney (Fig 3H). Multiple breast cancer patient-derived xenograft (PDX) specimens also clearly showed the existence of CASP7-p29/30 fragments (Fig 3I). Further, a panel of anthracycline-resistant MDA-MB-231 cell lines were analyzed, with the majority showing a clear accumulation of CASP7-p29/30 fragments (Fig 3J). Altogether, these data confirm the existence of stable non-canonical CASP7 fragments in vitro and in vivo models and in both normal and cancer tissues.

## CASP7-p29 and p30 are generated via processing at calpain cleavage sites

Next, we investigated the identity of the CASP7-p29/30 fragments observed in non-lethal stress conditions. In apoptosis, CASP7 activation is initiated by removing the pro-domain [37], and no stable accumulation of CASP7 lacking the pro-domain has been reported. To determine whether the non-canonical CASP7-p29/30 fragment(s) could be the result of pro-domain removal, we stably transfected CASP3 + 7 DKO cells with a full length CASP7

wild-type construct (CASP7-WT), a pro-domain deletion construct (CASP7-ΔPro) (Fig 4A and 4B), or a pro-domain cleavage site-mutant construct (S4A Fig). In western blot analyses, the CASP7-p29/30 fragments formed in starvation did not align with the CASP7-ΔPro construct, which appeared at ~ 33.5 kDa (Figs 4A and S4A). Additionally, when starved, the CASP7-ΔPro-expressing cells also resulted in CASP7-p29/30 fragment(s) (Figs 4A and S4A). Collectively, these data show that the non-lethal cellular stress-associated CASP7 cleavage occurs at a non-canonical site(s) downstream of the pro-domain cleavage site.

To elucidate the precise identity of the non-canonical cleavage sites and fragments, we enriched for CASP7-p29/30 fragments by performing CASP7-immunoprecipitation (IP) under starvation conditions (S4B). Protein in the 29/30 kDa region was extracted and subjected to Edman sequencing, which revealed the location of a non-lethal stress-associated cleavage site 10 amino acids downstream of the CASP7 pro-domain cleavage site (Fig 4B and 4C). This non-canonical cleavage site and a second site nearby (Fig 4B and 4C), corresponding to p30 and p29, respectively, were shown previously to be cleaved in vitro by calpains, but were associated with further processing of CASP7 to yield p17 and p18 [39,40]. Using the DeepCalpain algorithm [67], we identified predicted calpain cleavage sites conserved in murine CASP7 (S4C Fig) and confirmed that they corresponded to the size of the CASP7-p30/p29 bands detected in murine samples (Fig 3H). To verify the cleavage sites, we created a CASP7-calpain cleavage mutant (CASP7-CCM) construct by substituting both calpain cleavage sites with alanine residues (see Materials and methods; Fig 4C). Upon starvation, CASP3 + 7 DKO SKBR3 cells stably transfected with CASP7-CCM failed to form abundant CASP7-p29/p30 compared to CASP7-WT cells (Fig 4C). Together, these results validate the non-canonical cleavage of CASP7 and suggest a potential involvement of calpains in regulating CASP7 to promote starvation or PI-induced autophagy and/or cell survival in non-lethal stress conditions.

To determine whether calpains have any direct involvement in CASP7 non-canonical processing, we performed single KD of the most abundant calpain family members calpain 1 and calpain 2 in SKBR3 cell lines and subjected these cells to non-lethal amino acid starvation or PI stress conditions (Fig 4D–4I). Single KD of calpain 1 or calpain 2 led to a significant increase in overall CASP7 levels compared to control (Fig 4D–4I), suggesting a role in CASP7 turnover. Next, the ratio of CASP7 p29/30 fragment(s) relative to its full length was quantified (Fig 4G and 4I). In both stress conditions, calpain 1 KD cells showed the highest level of autophagy induction (Fig 4F and 4H) and CASP7-p29/30 fragments ratios (Fig 4F and 4H). In contrast, in both stress conditions, calpain 2 KD showed a reduction of stress-induced autophagy ($p < 0.05$ for PI-induced stress) and the lowest CASP7-p29/30 ratios. Collectively, these data suggest that calpain 2, but not calpain 1, may mediate the CASP7 non-canonical cleavage at the two calpain cleavage site(s) to positively regulate autophagy. If CASP3 is processed by a mechanism other than calpain 2, then CASP3 KO in combination with calpain 2 KD would be expected to result in an enhanced suppression of autophagy. However, we found that knockdown of calpain 2 in the CASP3 KO background did not further enhance autophagy inhibition relative to the scramble-siRNA control (S4D and S4E Fig), consistent with the possibility that CASP3 could also be processed by calpain 2 in this context. CASP3 was shown previously to be processed by calpains [68,69], further supporting the investigation of CASP3 processing in non-lethal stress conditions as an interesting avenue for future studies.

## PARylation and ATG gene expression are reduced in CASP3 and CASP7 DKO cells

The well-known PARP1 binding site [43] in CASP7 is located between the two identified calpain cleavage sites (S4F Fig). PARP1 and associated PARylation activities were shown to

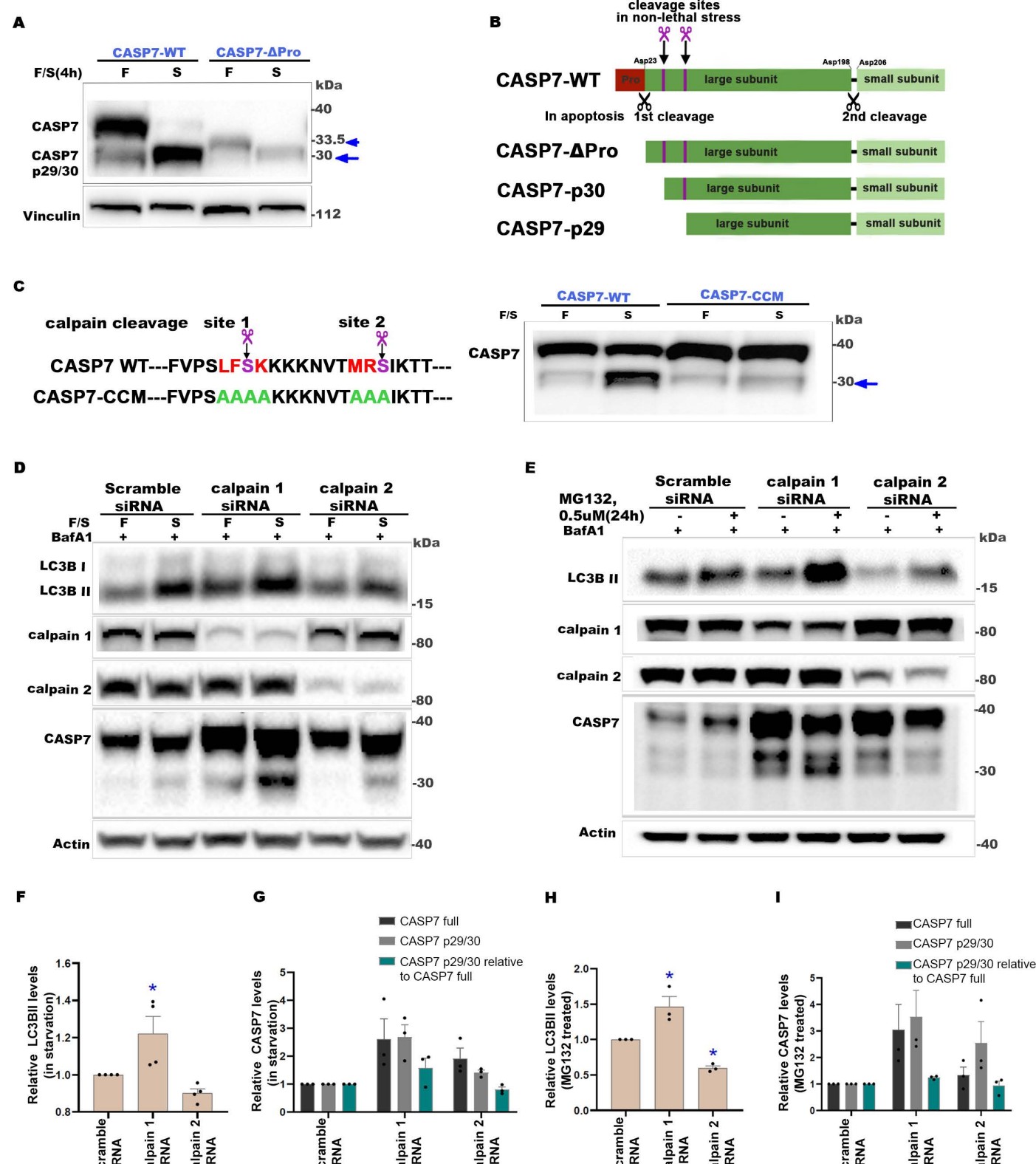

**Fig 4. CASP7-p29 and p30 are generated via processing at calpain cleavage sites.** (**A**) Representative western blot showing CASP7 immunolabeling in CASP3 + 7 DKO SKBR3 cells stably transfected with CASP7-WT or CASP7-delta-Pro (CASP7-Δpro) constructs in fed or starvation conditions. Arrowhead indicates CASP7-Δpro band and arrow indicates CASP7-p29/p30 bands. $n = 2$ independent experiments. (**B**) Schematic of subunits, canonical cleavage sites and

non-canonical cleavage sites in CASP7 wild-type or in various CASP7 fragments used or observed in this study. Black scissors denote the canonical cleavage sites and the order of processing in apoptosis. Purple scissors, bars, and black arrows denote the putative calpain cleavage sites (non-canonical). (**C**) The sequences on the left show the putative calpain cleavage sites in CASP7-WT and the amino acid replacements made in the CASP7 construct with putative calpain-cleavage sites mutated (CASP7-CCM). Representative western blot of CASP7 from CASP3 + 7 DKO SKBR3 cells transfected with CASP7-WT or CASP7-CCM constructs grown in fed (F) or starved (S) conditions. $n = 2$ independent experiments. (**D**, **E**) Representative western blots of indicated proteins from SKBR3 cells transfected with scramble, calpain 1 or calpain 2 siRNAs (48 h) and then continued to be cultured in fed (F) conditions or starved (S) for 8 h (D) or treated with MG132 for 24 h (E), with BafA1 (50 nM) in the final 2 h. (**F–I**) Quantification of LC3B-based autophagy flux and CASP7 bands shown in (D) and (E) relative to scramble-siRNA control. In graphs, data are shown as mean ± SEM. $n = 3$ independent experiments. *$P < 0.05$, **$P < 0.01$, ***$P < 0.001$, ****$P < 0.0001$, NS, not significant. See also S4 Fig. In F–I, one-way ANOVA with Dunnett's post-test. The numerical data presented in this figure can be found in S1 Data.

play key roles in stress sensing, stress adaptation, and autophagy induction in multiple cell lines and in vivo mouse models [70–75]. Further, PARP1 is a CASP3 and CASP7 substrate, with the latter showing a higher affinity for PARP1 binding [76]. These factors prompted us to investigate PARP1 levels and PARP1 activity in the CASP3 + 7 DKO background. We predicted that cleaved-PARP1 levels would be reduced, and thus PARP1 activity would be increased in the CASP3 + 7 DKO cells. Unexpectedly, we detected a higher level of cleaved-PARP1 in CASP3 + 7 DKO cells compared to the SKBR3 parental cells (Fig 5A–5D). There was no evidence of cell death in the DKO cells, consistent with previous observations that cleaved-PARP1 (89 kDa) is not always associated with cell death [77,78]. Cleaved-PARP1 levels were further increased when cells were stressed with a non-lethal dosage of MG132 (Fig 5B and 5D). PARylation is a PARP catalytic activity-dependent process [79]. Next, PAR immunolabeling was performed as a readout for PARylation activity of PARP1 [80]. We observed a significant reduction in PAR labeling in CASP3 + 7 DKO cells compared to parental cells in both SKBR3 and MDA-MB-231 cell lines (Fig 5E–5H). These results indicate that in the absence of CASP3 and CASP7, PARP1 cleavage is enhanced and PARP1 activity (as measured by PAR level) is impaired.

In cellular stress conditions, PARP1 and PARylation were shown previously to positively regulate cytoprotective autophagy through multiple mechanisms [73–75,81], including the transcriptional upregulation of autophagy pathway components) [82]. Since CASP3 + 7 DKO lead to reduced PARylation, we predicted that CASP3 + 7 DKO would result in transcriptional downregulation of autophagy pathway components. Among the components evaluated by reverse transcription polymerase chain reaction reverse transcription polymerase chain reaction (RT-qPCR), we found that *LC3B* and *ATG7* transcript levels were reduced in CASP3 + 7 DKO SKBR3 cells (S5A–S5E Fig). There was an increase in both transcripts in response to MG132-induced stress (Figs 5I and S5F), with DKO cells showing weaker upregulation compared to parental (Figs 5I and S5F). Dual re-expression of CASP3 + 7-WT constructs partially rescued transcript levels of these genes in DKO cells (Figs 5I and S5F). These results are consistent with the possibility that CASP3 and CASP7 regulate transcription of key autophagy genes, potentially through changes to PARP1 and PARylation.

To test whether CASP3 and CASP7 regulate autophagy through PARP1, we investigated whether overexpression of PARP1 could rescue autophagy in the CASP3 + 7 DKO background. Moreover, we compared rescue ability of wild-type PARP1 (PARP1-WT), catalytically inactive PARP1 (PARP1-CI), and PARP1 with the DEVD cleavage site mutated (PARP1-DEVA, PARP1-AEVA). Consistent with endogenous PARP1, the PARP1-WT transfected CASP3 + 7 DKO cells resulted in a significantly increased accumulation of cleaved-PARP1 compared to the PARP1-WT transfected parental cells (S5G and S5H Fig). As expected, levels of cleaved PARP1 were significantly reduced in the PARP1-DEVA and PARP1-AEVA expressing cells (S5G and S5H Fig). While the cleavage-resistant PARP1-DEVA and PARP1-AEVA

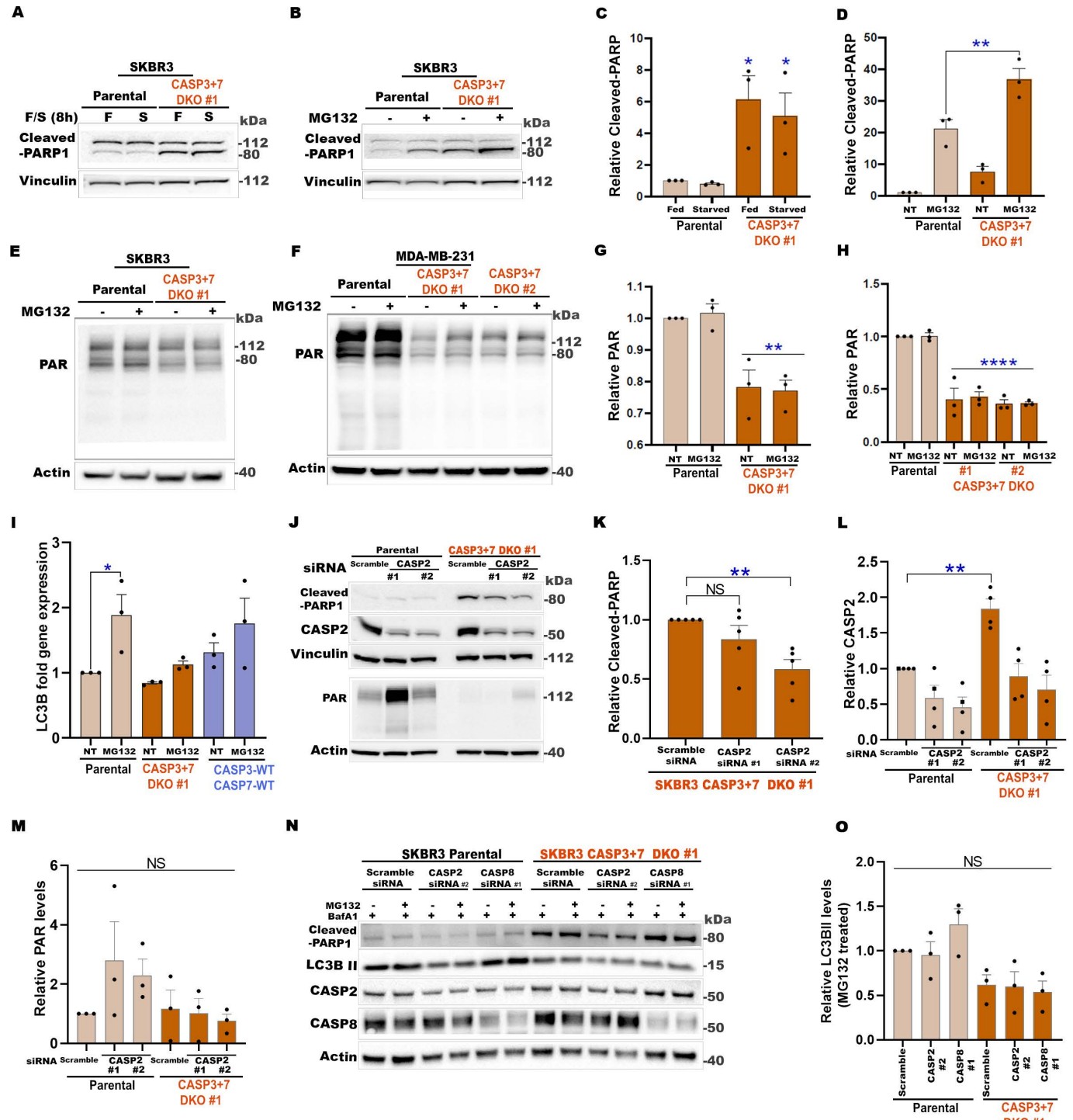

**Fig 5. PARylation and ATG gene expression are reduced in CASP3 +7 DKO cells.** (**A**, **B**) Representative western blots of indicated proteins from parental and CASP3 + 7 DKO SKBR3 cells cultured in fed conditions (F) or starved (S) for 8 h (A) or treated with MG132 (0.5 μM) for 24 h (B). (**C**) Quantification of cleaved-PARP1 shown in (A). Values were normalized to parental fed conditions. (D) Quantification of cleaved-PARP1 shown in (B). Values were normalized to parental NT. NT = No Treatment (vehicle DMSO). (**E**, **F**) Representative western blots of PAR immunolabeling from parental and CASP3 + 7 DKO SKBR3 (E) or MDA-MB-231(F) cells treated with vehicle (DMSO) or MG132 (0.5 μM) for 24 h. (**G**, -**H**) Quantification of PAR levels shown in (E) and (F). Values were normalized to the parental NT condition. (I) RT-qPCR analyses of LC3B in SKBR3 parental, CASP3 + 7 DKO or CASP3 + 7-WT re-expression in CASP3 + 7 DKO SKBR3 cells. Cells were treated with vehicle (DMSO) or 0.5 μM of MG132 for 24 h. (**J**) Representative western blots of indicated proteins from SKBR3 parental and CASP3 + 7 DKO cells transfected with scramble or CASP2 siRNAs (72 h). (**K**) Quantification of cleaved-PARP1 levels (shown in J) relative to scramble-siRNA control. (**L**)

Quantification of CASP2 levels relative to scramble-siRNA-treated parental control. (**M**) Quantification of PAR levels shown in (J) relative to scramble-siRNA-treated parental control. (**N**) Representative western blots of indicated proteins from SKBR3 parental and CASP3 + 7 DKO cells transfected with scramble, CASP2 or CASP8 siRNAs (42 h) and treated with vehicle DMSO or MG132 (0.5 μM) for 24 h, with BafA1 (50 nM) in the final 2 h. (**O**) Quantification of LC3B-based autophagy flux in MG132-treated cells, shown in (N). In graphs, data are shown as mean ± SEM. $n$ = 3–6 independent experiments. *$P < 0.05$, **$P < 0.01$, ***$P < 0.001$, ****$P < 0.0001$, NS, not significant. In C, D, G, and H, one-way ANOVA with Tukey's post-test. In I, K–M, and O, one-way ANOVA with Dunnett's post-test. See also S5 Fig. The numerical data presented in this figure can be found in S1 Data.

cells also showed a trend of increased LC3B transcription relative to PARP1-WT (S5J Fig), the PARylation and autophagy flux levels showed no differences (S5G, S5I, S5K, and S5L Fig). The PARP1-CI transfected cells did show significantly reduced PARylation compared to the vector control, consistent with the reported dominant negative effect of catalytically inactive PARP1 (S5G and S5I Fig) [83], but induction of cell death and very low expression levels confounded interpretation of results. Overall, while the association between PARP1 cleavage and LC3B transcription remains consistent, it is not clear whether the reduced autophagy flux in the CASP3 + 7 DKO background is due to the enhanced PARP1 cleavage.

Next, to identify the protease(s) responsible for PARP1 cleavage in the absence of CASP3 and 7, we evaluated multiple candidate proteases using a siRNA approach. The knockdown of CASP6, calpain 1, calpain 2, cathepsin B, and cathepsin D resulted in a further increase in PARP1 cleavage (S5M–S5O Fig). We next investigated caspase 2 (CASP2) and caspase 8 (CASP8), both shown to localize to the nucleus in some contexts [84,85]. We discovered that knockdown of CASP2, but not CASP8, consistently reduced PARP1 cleavage in CASP3 + 7 DKO cells (Figs 5J, 5K, and S5P). Interestingly, the relative expression level of CASP2 was significantly higher in CASP3 + 7 DKO cells compared to the parental cells (Fig 5J and 5L), further supporting a role for CASP2 in this context. Consistent with the lack of rescue by the DEVD cleavage site mutants, PARylation and autophagic flux were not rescued in the CASP2 knockdown CASP3 + 7 DKO cells (Fig 5M–5O). Collectively, our data suggest that CASP2 is at least partially responsible for the increased PARP1 cleavage in the CASP3 + 7 DKO cells. Further investigation of the contexts and role of CASP2 in PARP1 regulation is warranted.

## CASP7-p29/30 promote phosphorylation of DDR pathway protein H2AX

In addition to transcriptional remodeling, PARP1 plays a central role in DNA repair and DNA damage response (DDR) pathways in response to cell stress [71,86]. Both starvation and proteasome inhibition were reported to induce DNA damage and activate autophagy as a part of stress-induced DDR [87,88]. Western blot analyses revealed that MG132 stress-induced phospho-H2AX (γ-H2AX) accumulation was significantly reduced in CASP3 + 7 DKO cells compared to parental cells without reduction in total H2AX levels (Fig 6A and 6B).

Traditionally, the level of γ-H2AX was measured to determine the extent of DNA damage [89]; however, γ-H2AX levels were also shown to represent the strength of DDR signaling activity [12,90,91]. To distinguish between these two possibilities, we used comet assays to directly measure DNA damage [92]. The level of DNA damage/fragmentation in parental versus CASP3 + 7 DKO SKBR3 cells was not significantly different when treated with the vehicle (DMSO) control (Fig 6C and 6D). In response to non-lethal MG132 stress conditions, there was an increase in DNA damage, but it was comparable in both parental and CASP3 + 7 DKO cells. In contrast, the positive control, $H_2O_2$, lethal stress conditions induced an increase in DNA damage only in the parental cell line (Fig 6C and 6D). This data indicates that the γ-H2AX reduction observed in CASP3 + 7 DKO cells in the MG132 non-lethal stress

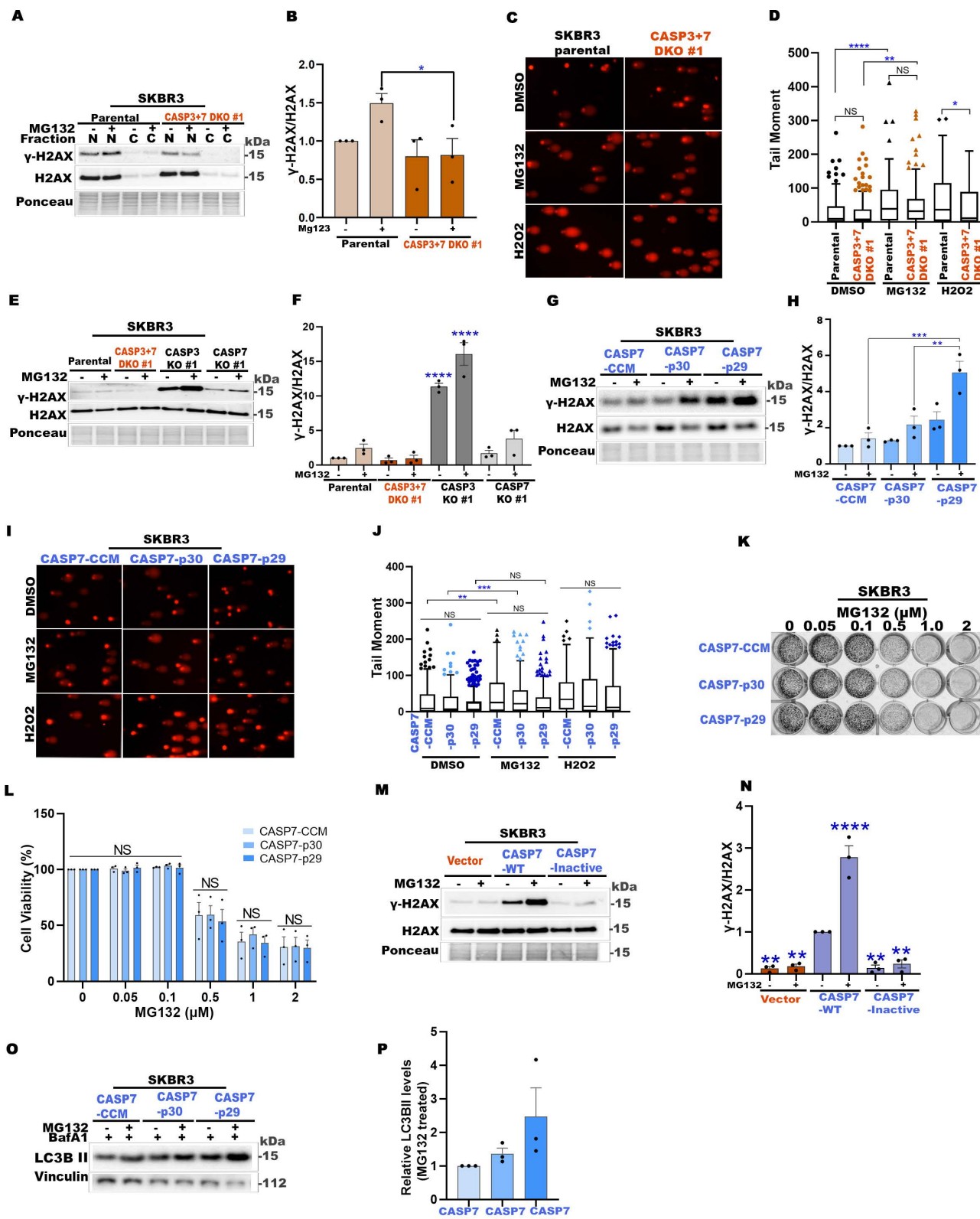

**Fig 6. CASP7-p29/30 promote phosphorylation of DDR pathway protein H2AX. (A)** Representative western blots of indicated proteins from SKBR3 parental and CASP3 + 7 DKO cells treated with vehicle DMSO or MG132 (0.5 µM) for 24 h. Ponceau S staining was used as the loading control. **(B)** Quantification of γ-H2AX/total H2AX shown in **(A)**. **(C)** Representative images from alkaline comet assay from SKBR3 parental and

CASP3 + 7 DKO cells treated with vehicle DMSO or MG132 (0.5 μM) for 24 h. Cells treated with $H_2O_2$ (100 μM for 4 h) serve as the positive control. **(D)** Quantification of tail moments of comets shown in **(C)**. Tukey's box-and whisker plots are based on tail moments determined by CometScore from 200 cells in each of two independent assays. **(E)** Representative western blots of indicated proteins from SKBR3 parental, CASP3 + 7 DKO or single KO cells treated with vehicle DMSO or MG132 (0.5 μM) for 24 h. Ponceau S staining was used as the loading control. **(F)** Quantification of γ-H2AX/total H2AX shown in **(E)**. **(G)** Representative western blots of indicated proteins from CASP3 + 7 DKO SKBR3 cells stably transfected with indicated CASP7 constructs treated with vehicle DMSO or MG132 (0.5 μM) for 24 h. Ponceau S staining was used as the loading control. **(H)** Quantification of γ-H2AX/total H2AX shown in **(G)**. **(I)** Representative images from alkaline comet assay from CASP3 + 7 DKO SKBR3 cells stably transfected with indicated CASP7 constructs and treated with vehicle DMSO or MG132 (0.5 μM) for 24 h. Cells treated with $H_2O_2$ (100 μM for 4 h) serve as the positive control. **(J)** Quantification of tail moments of comets shown in **(I)**. Tukey's box-and-whisker plots are based on tail moments obtained from 200 cells in each of two independent assays. **(K)** Representative images of crystal violet assay plate. CASP7-CCM, CASP7-p30, and CASP7-p29 SKBR3 cells treated with indicated concentrations of proteasome inhibitor MG132 for 24 h and continued to grow in drug free media for another 3 days. **(L)** Quantification of cell viability data presented in **(K)**. The percentage of stained (viable) cells at each concentration was normalized to respective untreated cells. **(M)** Representative western blots of indicated proteins from CASP3 + 7 DKO SKBR3 cells re-expressing vector, CASP7-WT (catalytically active) or CASP7-inactive (catalytically inactive) constructs, treated with vehicle DMSO or MG132 (0.5 μM) for 24 h. H3 serves as the loading control. **(N)** Quantification of γ-H2AX/total H2AX shown in **(M)**. **(O)** Representative western blots of LC3B-based autophagy flux in CASP3 + 7 DKO SKBR3 cells re-expressing the indicated CASP7 constructs, treated with vehicle DMSO or MG132 (0.5 μM) for 24 h, with BafA1 (50 nM) in the final 2 h. **(P)** Quantification of autophagy flux in MG132 treated cells relative to MG132 treated CCM expressing cells, shown in **(O)**. In graphs, data are shown as mean ± SEM. $n = 3$ independent experiments except for comet assays $n = 2$. *$P < 0.05$, **$P < 0.01$, ***$P < 0.001$, ****$P < 0.0001$, NS, not significant. In B, F, L, N, and P, one-way ANOVA with Dunnett's post-test. In D and H, one-way ANOVA with Tukey's post-test. In J, two-way ANOVA with Sidak's post-test. See also S6 Fig. The numerical data presented in this figure can be found in S1 Data.

conditions is not due to a reduction in DNA damage, but possibly instead due to impaired DDR signaling.

Next, we investigated the γ-H2AX levels in CASP3 + 7 double and single KO cells (Fig 6E and 6F). The γ-H2AX levels were reduced in CASP3 + 7 DKO cells, but relatively increased in both CASP3 single KO and CASP7 single KO cells. Notably, the γ-H2AX level was highest in CASP3 KO cells, suggesting that CASP7 plays a prominent role, relative to CASP3, in the H2AX phosphorylation. To further investigate this possibility and given the proximity of the PARP1 binding site to the CASP7 calpain cleavage sites, we generated CASP3 + 7 DKO cell lines stably expressing CASP7-CCM, CASP7-p30, or CASP7-p29 (S6A Fig). We used CASP7-CCM as the full-length control since it does not undergo non-canonical cleavage. We compared γ-H2AX levels in the absence and presence of non-lethal MG132, and observed that the stress-induced γ-H2AX accumulation was increased in CASP7-p30 and CASP7-p29 expressing cells compared to CASP7-CCM expressing cells (Fig 6G and 6H). Among the three constructs, the greatest level of γ-H2AX was observed with CASP7-p29. Furthermore, comet assays revealed that the degree of DNA damage was not significantly different among the three cell lines in either vehicle (DMSO) control, non-lethal MG132-treated or lethal $H_2O_2$-treated conditions (Fig 6I and 6J). In fact, the degree of DNA damage appeared slightly reduced in CASP7-p29 expressing cells, where we detected the greatest level of γ-H2AX (Fig 6H and 6J). Crystal violet cell viability assays, following treatment with increasing concentrations of MG132 or Bortezomib (Figs 6K, 6L, S6B and S6C), showed no significant differences in cell viability despite the different γ-H2AX levels observed in the three cell lines. Overall, these results support that CASP7 non-canonical processing does not affect the degree of DNA damage, but rather plays a role in inducing different levels of γ-H2AX signaling (S6D Fig). Consistent with several previous studies showing that γ-H2AX is not an unambiguous marker for DNA double-strand breaks [12,90,91], our findings emphasize the need for careful interpretation of γ-H2AX levels.

To further explore CASP7 in this role, we employed catalytically active wild type (CASP7-WT) and catalytically inactive (CASP7-inactive, C > A) constructs stably expressed in CASP3 + 7 DKO cells. CASP7-WT construct expression, but not catalytically inactive construct, led to a significant increase in the γ-H2AX signaling in non-lethal MG132-treated cells

(Fig 6M and 6N), revealing that CASP7 and its catalytic activity are required for the phosphorylation of H2AX.

## Dual inhibition of CASP3 and CASP7 mimics effects of PARP1 inhibitors

Our data shows that inhibition of CASP3 and CASP7 leads to increased PARP1 cleavage and reduced PARP activity (PARylation), DDR signaling, and autophagy response. Since these effects phenocopy the genetic or pharmacological inactivation of PARP1 [70,71], we hypothesized that dual CASP3 + 7 inhibition would also phenocopy the synthetic lethal effects of PARP1 in cells that are defective in homologous recombination (HR) [93,94]. To test this, we generated CASP3 + 7 DKD in the BRCA1-deficient SUM149PT breast cancer cells and assessed their viability by the crystal violet assay. In striking contrast to the BRCA1/2-proficient SKBR3, MDA-MB231, and JIMT1 breast cancer cells, the dual inhibition of CASP3 + 7 was lethal in SUM149PT cells (Fig 7A–7C). In the western blot analyses, a distinct band of cleaved-PARP1 was visible in CASP3 + 7 DKD cells compared to the scramble-siRNA control (Fig 7C). Western blot analyses also confirmed that autophagic flux was reduced in the CASP3 + 7 DKD SUM149PT cells compared to the scramble-siRNA control or compared to the single KD of CASP3 or CASP7 (Fig 7D and 7E). To determine whether the lethal effects of CASP3 + 7 DKD was redundant or potentially additive with PARP1 inhibition, we treated the SUM149PT cells with the PARP1 inhibitor olaparib. The combination of CASP3 + 7 DKD and olaparib treatment resulted in a further reduction of SUM149PT cell viability compared to olaparib treatment alone (Fig 7F and 7G). These results suggest that combined inhibition of CASP3 and CASP7 induces synthetic lethality with loss of BRCA1.

To further assess genetic interactions between BRCA1 and CASP3 and CASP7, we utilized Genetic inteRaction and EssenTiality neTwork mApper (GRETTA) [95], a tool that leverages the publicly available data from the Cancer Dependency Map (DepMap [96–98], to perform a pan-cancer *in silico* genetic interaction screen. Using GRETTA, we identified 16 DepMap cancer cell lines harboring low protein expression of both CASP3 and CASP7 and 10 cancer cell lines with high expression of both CASP3 and CASP7 across 13 cancer types (Figs 7H and S7A). Comparisons between these two groups resulted in the identification of 142 candidate CASP3 + 7 synthetic lethal interactors, where perturbation of these genes resulted in significantly higher lethal probabilities in the CASP3 + 7 low expressor lines compared to the CASP3 + 7 high expressor lines (S7B Fig). The candidate synthetic lethal interactors included HR repair pathway genes (S7B Fig), with BRCA1 as the top synthetic lethal hit (Fig 7I). Collectively, our results show that the dual inhibition of CASP3 and CASP7 mimics the cellular effects of PARP1 inhibitors, revealing a surprising new avenue for potential therapeutic exploitation. Counter-intuitive to traditional caspase roles, our findings demonstrate that combined inhibition of CASP3 and CASP7 induces synthetic lethality in BRCA1-defective cells.

## Discussion

Caspases remain categorized primarily as components of the cell death machinery despite studies demonstrating that they have multiple non-apoptotic functions. We discovered an autophagy-promoting and cell stress adaptation role for the human effector caspases, CASP3 and CASP7, in human breast cancer cell lines in non-lethal starvation or proteasome inhibition stress conditions. Using non-lethal stress conditions, we found that CASP7 is cleaved, resulting in stable fragments (CASP7-p29/30) that are distinct from canonical cleaved-CASP7 fragments (CASP7-p20/p12) reported in apoptosis. The non-canonical cleavage sites in CASP7 flank a PARP1 binding site, and we found that PARP activity and DNA damage-induced phosphorylation of H2AX are significantly reduced in CASP3 + 7 DKO cells.

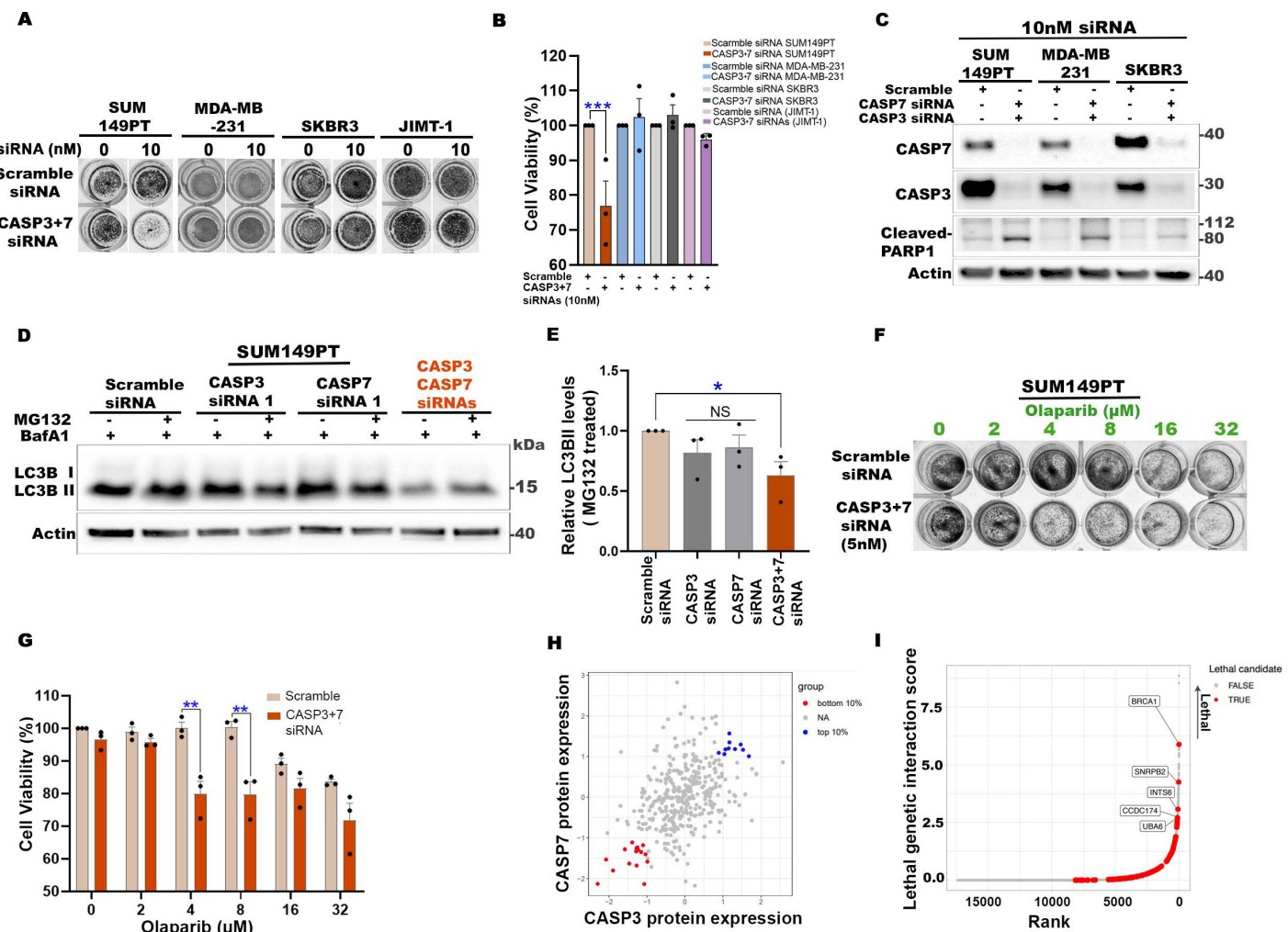

**Fig 7. Dual inhibition of CASP3 and 7 mimics effects of PARP1 inhibitors.** (**A**) Representative images of crystal violet assays. SUM149PT, MDA-MB-231, SKBR3, or JIMT-1 cells transfected with scramble-siRNA or CASP3 and CASP7 siRNAs (10 nM) for 2 days and then cultured in siRNA-free media for another 3 days. (**B**) Quantification of cell viability from data represented in (**A**). Percentage of stained (viable) cells at each concentration was normalized to respective untreated cells. (**C**) Representative western blots showing CASP3 and CASP7 levels following siRNA treatment conditions used in crystal violet assays shown in (**A**). Immunolabelling of cleaved-PARP1 is also shown. $n = 2$ independent experiments. (**D**) Representative western blots of indicated proteins from SUM149PT cells transfected with scramble, CASP3 and/or CASP7 siRNAs (48 h) and treated with vehicle DMSO or MG132 (0.5 μM) for 24 h, with BafA1 (50 nM) in the final 2 h. (**E**) Quantification of LC3B-based autophagy flux in proteasome inhibitor (MG132) treated cells shown in (**D**). The levels of LC3BII in MG132 treated cells were normalized to loading control and shown relative to the MG132 treated scramble-siRNA control. (**F**, **G**) Representative images of crystal violet assay plates (F) and quantification of percentage of cell viability (**G**). Indicated siRNA transfected cells treated with indicated concentrations of Olaparib for 24 h and continued to be cultured in drug free media for another 3 days. Graph (G) show the percentage of stained (viable) cells at each concentration normalized to scramble-siRNA-treated and Olaparib untreated cells. (**H**) Dot plot showing DepMap cancer cell lines with bottom 10% of CASP3 + 7 protein expression ($n = 16$) and top 10% CASP3 + 7 protein expression ($n = 10$). Normalized CASP3/7 protein expression (scaled and mean centered) were extracted from Nusinow and colleagues using GRETTA. (**I**) Ranked CASP3 + 7 genetic interaction scores generated using GRETTA. The cell lines at the bottom 10% of CASP3 and CASP7 protein expression are compared against cells lines at the top 10%. Rank is based on lethal GI scores. The red points indicate true candidate CASP3 + 7 lethal genetic interactions (GIs). The top most lethal genetic interactors are labeled. In graphs, unless otherwise noted, data are shown as mean ± SEM. $n = 3$ independent experiments. \*$P < 0.05$, \*\*$P < 0.01$, \*\*\*$P < 0.001$, \*\*\*\*$P < 0.0001$, NS, not significant. In B and E, one-way ANOVA with Sidak's and Dunnett's post-test, respectively. In G, two-way ANOVA with Sidak's post-test. In I, Mann–Whitney $U$-test $p$-value < 0.05. See also S7 Fig. The numerical data presented in this figure can be found in S1 Data. The code related to Fig 7H and 7I is publicly available in a GitHub repository (https://github.com/MarraLab/Caspase_GRETTA_analysis) and archived on Zenodo (https://doi.org/10.5281/zenodo.14722298).

Strikingly, BRCA1-deficient cells that are known to be highly dependent on PARP activity for DNA repair, exhibited synthetic lethality with combined knockdown of CASP3 + 7. Overall, this study shifts the current paradigm of apoptotic caspase-PARP1 relationships to one that involves non-canonical caspase cleavage and the promotion of cell adaptation and survival pathways at the onset of cellular stress or in non-lethal cellular stress conditions.

Unexpected key findings of our study include the increased PARP1 cleavage and decreased PARylation (PARP activity) in CASP3 + 7 DKO cells. There is extensive evidence that CASP3 and CASP7 function as negative regulators of PARP1, cleaving and inactivating it during apoptosis [99–101]. However, our data suggest that CASP3 and CASP7 alternatively contribute to PARP1 stability and function in non-lethal conditions. Our findings, along with the Conde-Rubio and colleagues (2021) [7] report of differential cleavage of PARP1 by CASP3 and CASP7 in non-lethal stress as compared to lethal stress, suggest that the primary outcome of the CASP-PARP1 relationship during non-lethal stress is not to inactivate PARP1 activity, but rather is to modulate PARP1 activity. We identified a role for CASP2 in PARP1 cleavage in the absence of CASP3 and CASP7, but the upstream signals remain to be identified.

Given the location of the PARP binding site on CASP7 (Fig 4C), it is likely that PARP cleavage, stability and/or activity are modulated by non-canonical cleavage. This N-terminus 'KKKK' exosite located downstream of the first calpain cleavage site on CASP7 has been shown to promote efficient PARP1 binding and cleavage [43]. The exosite is directly exposed for PARP1 binding in CASP7-p30. It has been shown that removal of the pro-domain enhances the PARP1 binding potential of CASP7 [43]. However, it is not known whether the direct exposure of the exosite would further enhance or instead abrogate the binding or cleavage of PARP1. Since CASP7-p29 does not have an intact exosite, we initially expected it to differently modulate PARP1 and downstream stress response pathways. Despite this key difference, expression of either exogenous p29 or p30 alone was able to enhance the autophagy response or the DDR (γ-H2AX), revealing a potential functional similarity, which is likely independent of PARP1 binding, between these non-canonical CASP7 fragments. However, the p29 fragment lacking the PARP1 binding exosite shows the significantly highest level of LC3B-II flux or γ-H2AX. This result is somewhat consistent with an in vitro cell-free analysis that demonstrated higher enzymatic activity when CASP7 cleavage occurs at the second calpain cleavage site (equivalent to the p29 site), compared to the p30 or canonical cleavage sites [40]. In that study; however, the initial processing was followed by a second cleavage event to generate the enzymatically active tetrameric complex composed of p17 and p12 fragments [40]. In non-lethal contexts, we predict that the retention of CASP7-p30 and/or p29 forms serves to limit apoptotic activity. A model to be investigated in the future is whether p30 initiates the stress response signaling, but then switches to p29 to maximize the ability to respond to increased levels of stress. Despite the differences in autophagy or γ-H2AX levels, we did not observe differential effects on cell viability in lines expressing the different CASP7 fragments under normal or PI stress conditions. Several studies have reported non-lethal functions of caspases in DDR as well as in non-lethal DNA damage in chromatin modulation leading to changes in gene expression [3,102,103]). An increase in γ-H2AX levels was also reported to be caspase-dependent and associated with increased carcinogenesis or cell invasion [12,104]. It is possible that the non-canonical processing of CASP7 may have physiological effects on adaptations and/or survival that cannot be captured within our experimental window. Notably, since we did not detect further processing of p29/p30, it appears that these fragments function, at least in part, without formation of the traditional tetrameric complex. Future studies are needed to understand the structural and functional properties of CASP7-p29 and p30.

PARP1 and PARylation have been well recognized as cellular hubs that modulate major stress response pathways [74,105]. It is possible that the impaired PARP1 activity in CASP3 + 7

DKO cells is responsible for the observed phenotypes, including the reduction in autophagy response. PARP1 and PARylation were shown previously to play direct and indirect roles in the upregulation of cytoprotective autophagy through multiple mechanisms, including modulation of AMPK and ATP levels [74,106], regulation of DDR signaling [87,88] or transcriptional upregulation of autophagy pathway components [75,81,82]. For example, LC3, GABARAPL1, ATG5, and ATG12 transcripts and proteins were shown to be upregulated by PARP1 in multiple cell types [71,82]. Somewhat consistent with these studies, we found that both ATG7 transcript and LC3B transcript and protein levels were significantly reduced in CASP3 + 7 DKO cells, which exhibited reduced PARylation. Cleavage-resistant PARP1-DEVA and PARP1-AEVA expressing cells exhibited a trend of increased LC3B transcript levels relative to PARP1-WT cells, but there was no apparent rescue of autophagy flux or PARylation phenotypes. However, expression level differences between constructs and the induction of cell death by the PARP expression constructs hindered direct comparisons of these readouts. Interestingly, CASP2 protein levels were increased in the CASP3 + 7 DKO cells and CASP2 KD in this background resulted in a modest, but consistent reduction in cleaved PARP levels. We did not detect a significant rescue of PARylation or autophagy flux following CASP2 KD, but further studies using alternate approaches to enhance KD or KO CASP2 are warranted. Additional investigations are required to define the precise relationship between PARP1 and autophagy in the presence and absence of CASP3 and CASP7.

A limitation of our study relates to the potential non-canonical processing of CASP3 and the relative contributions of CASP3 and CASP7 in autophagy and PARP1 modulation. We did detect a non-canonical CASP3 fragment of 27 kDa in non-lethal stress conditions (Fig 3D). Although this observation was not consistent, it would be valuable to further investigate the nature and stability of this CASP3 fragment. In accordance with our findings that CASP3 plays a role in autophagy upregulation, the pro-domain of CASP3 has been shown to be involved in the clearance of intracellular protein aggregates [107–109]. Liu and colleagues (2015) [12] compared the relative contribution of CASP3 and CASP7 in γ-H2AX foci formation in WT, CASP3KO, CASP7KO, and CASP3 + 7 DKO MCF-10A cells and concluded that CASP3 and CASP7 have largely overlapping functions in terms of γ-H2AX foci formation, but with CASP3 playing a more dominant role. Our study is consistent with an overlapping role for both CASP3 and CASP7 in increasing the level of γ-H2AX, but induction of H2AX phosphorylation with CASP7-p29/30 suggests that non-canonical forms of CASP7 may also have a prominent role in this process. Further, the catalytically inactive form of CASP7 could not induce the phosphorylation of H2AX showing that the catalytic activity of CASP7 is required.

Our study uncovers a novel mechanism of stress adaptation used by cancer cells, revealing new vulnerabilities that can potentially be exploited as therapeutic targets or biomarkers. Counter-intuitively, our data supports the idea that caspase inhibition, rather than caspase induction, may reduce cell fitness in certain cells and stress contexts. Strikingly, the synthetic lethal phenotype we observed in BRCA1 deficient cells points to a unique context where CASP3 + 7 inhibition may be most effective. Based on our *in silico* findings, we expect that this synthetic lethal relationship will be relevant to multiple cancer cell types, and that CASP3 + 7 inhibition has synthetic lethal potential with other genes involved in HR repair or other processes (S7B Fig). Currently, PARP1 inhibitors are used clinically for the treatment of BRCA-mutated breast, ovarian, pancreatic, and prostate cancers [110,111]. CASP3 + 7 inhibition could be further explored as a potential alternative to PARP1 inhibitors or as a second-line treatment in cases of intrinsic or acquired resistance to PARP1 inhibitors. In addition, combining CASP3 + 7 inhibition with chemotherapeutic drugs that induce DNA damage (i.e., cisplatin, doxorubicin) might increase the drug sensitivity. Our study also suggests CASP3 + 7 inhibition as an alternate way to inhibit the autophagy pathway which

may be beneficial due to clinical challenges associated with existing autophagy inhibitors like hydroxychloroquine. While the safety and efficacy of CASP3 + 7 inhibition strategies [112,113] remain to be determined in these contexts, our study has identified multiple possibilities worthy of investigation.

An evolutionarily conserved role for caspases in stress adaptation was postulated previously [1,6]. In support of such a role for human CASP3 and CASP7, we have identified a molecular mechanism that involves modulation of PARP activity, phosphorylation of H2AX for DNA damage signaling, and autophagy. Stress-induced cytoprotective autophagy induction by a *Drosophila* effector caspase was previously linked to a mechanism involving regulation of ATP levels [35]. Notably, like caspases or the yeast metacaspase, phylogenetic studies suggest that the ancestor of all extant eukaryotes expressed ancestral PARP proteins, including one similar to human PARP1 [114]. It will be interesting to determine if stress-adaptive caspase modulation of PARP also occurs in *Drosophila* or other organisms. We predict that this will be the case and, given the observation of CASP7-p29/p30 fragments in multiple tissues from healthy mice, we also predict the results reported herein to have broad biological relevance. The non-lethal stress adaptation role for caspases identified in this study also provides a plausible mechanistic explanation for previous reports of associations between increased caspase expression and worse cancer patient outcomes. We propose this increased caspase expression is used to support tumor stress adaptation, through enhanced PARP1 activity, autophagy, and DDR signaling, and may be a targetable vulnerability, particularly in the context of BRCA1 or other potential synthetic lethal mutations identified here. In this regard, it will be important to determine whether blocking the CASP7 calpain cleavage site(s) or exosite instead of inhibiting entire CASP7 function is more beneficial in a therapeutic context. A detailed understanding of caspase function, mechanisms, and pathways in non-lethal stress will be crucial for the successful clinical translation of caspase-based inhibitors for the treatment of cancer. Our study provides unexpected molecular mechanisms and many avenues to further explore in this regard.

## Materials and methods

### Ethics statement

All experiments involving animals were conducted in accordance with the standards and guidelines of the Canadian Council on Animal Care and approval was granted by the University of British Columbia Animal Care Committee (A22-0274). Ethical approval (H18-02121, H23-01014) for research using human breast tumor tissues was granted by the Research Ethics Board of BC Cancer and Simon Fraser University.

### Cell lines and cell culture conditions

SKBR3, MDA-MB-231, and JIMT-1 breast adenocarcinoma cell lines (Parental, CASP KO, KD, or expressing CASP constructs) were maintained in Dulbecco's Modified Eagle Medium (DMEM) (Gibco, 11995-065) supplemented with 10% heat-inactivated fetal bovine serum (FBS) (Invitrogen, 12483020), 10 mM HEPES (Invitrogen, 15630–080), and 1× non-essential amino acids (Gibco, 11140-050). BRCA1 deficient SUM149PT breast cancer cells (parental or CASP KD) were maintained in Ham's F-12 medium (Gibco, 11765-054) supplemented with 5% heat-inactivated FBS (Invitrogen, 12,483,020), Insulin (1 µg/ml; Gibco, 12585-014), and hydrocortisone (1 µg/ml; Sigma, H4001). Cell lines stably transfected with plasmids expressing CASP constructs were maintained using Geneticin (G418) (Invitrogen, 10131-035) at 1 mg/ml or Hygromycin (Invitrogen, 10687010) at 500 µg/ml for selection. The selection media was replaced with standard media two passages before cells were used in experiments. All cells

were maintained at 37 °C with 5% $CO_2$ and 95% humidity. Testing for mycoplasma contamination of cell lines was conducted on a regular basis using e-Myco Mycoplasma Detection Kit (Boca Scientific, 25,235).

## Mice and PDX

To obtain cell lysate from murine organs, tissues from three female CD-1 mice, 10-weeks old, were flash-frozen and gifted by Dr. Nancy dos Santos and Nicole Wretham. All experiments were conducted in accordance with the standards and guidelines of the Canadian Council on Animal Care and were approved by the University of British Columbia Animal Care Committee (A22-0274).

To obtain cell lysate from PDX, breast tumor tissues originally derived from human were grown as xenografts in mice as detailed in Eirew and colleagues (2022) [115] harvested, frozen in DMEM/FCS with 6%–10% DMSO and provided by Dr.Sam Aparicio and Dr. Peter Eirew. Ethical approval for research using these human breast tumor tissues was obtained from the Research Ethics Board of BC Cancer and Simon Fraser University.

## Starvation, pharmacological, or Bafilomycin (BafA1) treatment

After growing for 3–4 days, cells were subjected to starvation or treated with proteasome inhibitors (or other drug treatments used in this study; rapamycin, staurosporine, olaparib). Cells were washed with 1× phosphate-buffered saline (PBS) (Glibco,10010023) and subjected to amino acid starvation for 2–24 h for autophagy flux assays or for 8–72 h for viability assays (as indicated) in Earle's Balanced Salt Solution (EBSS) (Sigma, E3024). For well-fed cells, the media was replaced with fresh standard growth media. For the autophagic flux assay with Bafilomycin A1 (BafA1) (Sigma, B1793), cells were incubated in 50 nM of BafA1 to inhibit lysosomal fusion for 2 h prior to harvesting cells for subsequent analyses. Staurosporine (Sigma, S6942), Rapamycin (Invitogen, PHZ1235), PARP inhibitor, Olaparib (Selleckchem, S1060), and proteasome inhibitors MG132 (Apexbio, A2585) and Bortezomib (Apexbio, A2614) stock solutions were prepared in dimethylsulfoxide (DMSO) (Fisher BioReagents, BP231-100) and stored in aliquots at − 20 °C. Drugs were freshly diluted in media to minimize DMSO concentration and cells were treated at indicated concentrations and durations. Control cells (non-treated or DMSO) were treated with a comparable concentration of DMSO.

## Cell lysate extraction for western blotting

Cells were plated in 6-well plates at a density of $2 \times 10^5$ cells per well for all cell lines except for MDA-MB-231 which was plated at $1.5 \times 10^5$ cells per well. For cell harvesting after indicated treatments (drugs or fed/starve conditions), cells were washed twice in PBS and then trypsinized for up to 5 min while in a $CO_2$ incubator. Cell pellets were collected by adding PBS and then spinning for 30 s, followed by rinsing with PBS and stored at −80 °C or processed to the next step. All cell pellets were resuspended and lysed using 3× packed cell volume of ice-cold RIPA lysis buffer (Santa-Cruz, SC24948) supplemented with sodium orthovanadate (Santa-Cruz, SC24948), protease inhibitor cocktail (Santa-Cruz, SC24948), and phenylmethylsulfonyl fluoride (Santa-Cruz, SC24948) following manufacturer's instructions. Cells were incubated in RIPA for 1 h at 4 00B0C with agitation before being centrifuged at 13,000 rpm for 10 min at 4 °C to obtain a lysate free of cell debris.

To prepare lysate from frozen mouse organs or PDXs, samples were cut into small pieces with a single-edge blade on a clean petri dish, transferred to lysis Matrix M tubes containing a single zirconium ceramic bead and RIPA lysis buffer (300–500 μl) supplemented with 1× complete-protease inhibitor (Roche) and 1× PhosSTOP (Roche, 4906845001) and incubated on

ice for 5 min. Samples were homogenized using MP Fast Prep Tissue Homogenizer (Biomedicals 116004500), spun down at 13,000 rpm for 2 min for debris to form pellets and then the supernatants were transferred to eppendorf tubes before being centrifuged at 13,000 rpm for 20 min at 4 °C. Lysates were transferred to new tubes avoiding pellets and floating lipid/fat layer.

### Western blot analysis

Proteins in cell/tissue lysates were quantitated using the Pierce BCA Protein Assay Kit (Invitrogen, 23225). For gel electrophoresis, 10–20 μg protein was prepared by adjusting the total volume using dH$_2$O and 4× SDS Loading buffer. Proteins were denatured by boiling at 95 °C for 10 min and then separated by gel electrophoresis (SDS PAGE) on a 4%–12% or 10% Bolt Bis–Tris Plus gel (Invitrogen NW04125BOX or NW00105BOX) with 1× NuPAGE MES SDS Running Buffer (Invitrogen, NP0002 at 180V for 40 min). Separated proteins were transferred to methanol-activated polyvinylidene difluoride (PVDF) membranes (BioRad, 1620177) at 100 V for 90 min in 1× Nupage Transfer Buffer (Invitrogen NP0006) with 20% (v/v) methanol. Membranes were blocked in 2% (w/v) skim milk in 1× PBS-T (PBS, 0.1% Tween-20, pH7.4) for 1 h at room temperature (RT), and incubated with primary antibodies overnight at 4 °C at 1:500 or 1:1000 dilution (see S3 Table for antibodies used) in Odyssey (Li-COR , 927-60003) plus 1× PBS-T (0.1%) solution mixed at 1:4 ratio.

The following day, membranes were washed three times, for 15 min each, with 1× PBS-T, blocked in 2% skim milk for 30 min at RT and incubated with HRP-conjugated corresponding (goat anti-mouse or goat anti-rabbit) secondary antibodies at 1:5,000 dilution in 1× PBS-T (0.1%) for 1 h at RT. After washing membranes with 1× PBS-T (0.1%) three times (5 min each), protein signals were detected with the Bio-Rad Clarity Western ECL Enhanced Western Blotting substrate (BioRad,170-5061) or with SuperSignal West Femto Maximum Sensitivity Substrate (Thermo Fisher Scientific, 34094) using Bio-Rad ChemiDoc XRS (or MP) System. Densitometry was performed using Image Lab 5.1 software (BioRad). Protein levels were normalized to the loading control, either β-actin or vinculin.

### Cell fractionation

Nuclear and cytoplasmic extracts for western blot analyses for H2AX and ɤ-H2AX were prepared using NE-PER Nuclear Cytoplasmic Extraction Reagent kit (Pierce) following the manufacturer's instructions. Briefly, harvested cells were washed twice with PBS and were centrifuged at 500 $g$ for 5 min. The cell pellet (packed volume of 20 μl) was suspended in 200 μl of cytoplasmic extraction reagent I, vortexed for 15 s, and incubated on ice for 10 min followed by the addition of 11 μl of ice-cold cytoplasmic extraction reagent II, vortexed for 5 s, incubated on ice for 1 min, and centrifuged for 5 min at 16,000 $g$. The supernatant (cytoplasmic extract) was transferred to a pre-chilled tube. The insoluble pellet fraction (nuclei), was resuspended in 100 μl of nuclear extraction reagent, vortexed for 15 s and incubated on ice for 10 min, before being centrifuged for 10 min at 16,000 $g$. The supernatant (nuclear extract) was transferred to a pre-chilled tube.

### Proteasome activity assay

Two thousand cells were seeded in a 96-well, white cell culture clear-bottom microplate (Greiner bio-one, 655983) and cultured in appropriate media (as indicated above) for up to 4 days. After treating cells with proteasome inhibitors for 24 h at indicated concentrations, proteasome activity was measured using Proteasome-Glo Cell-Based Reagents as instructed (Promega, G862112). Chymotrypsin-like, caspase-like, and trypsin-like activities were individually measured after incubating the corresponding luminogenic substrates with cells for

10 min. The degree of proteasomal activity (luminescence) was measured on a Synergy H4 Hybrid (BioTek) plate reader.

### siRNA transfection

For siRNA transfections, SKBR3 and SUM149PT cells were plated at $2 \times 10^5$ cells per well and MDA-MB-231 cells were plated at $1.5 \times 10^5$ cells per well, in 6-well plates in a total of 2 ml standard media. Twenty-four hours after plating, cells were transfected with 10 nM of CASP siRNA, calpain siRNA, cathepsin siRNA, or corresponding Scramble control siRNA (Integrated DNA Technologies for CASP-siRNAs and cathepsin-siRNAs and Santa-Cruz for calpain-siRNAs) using 4 μl of Lipofectamine RNAiMAX (Invitrogen, 13778075) as per manufacturer's recommendations. siRNAs are listed in S1 Table. 48 h after transfection, cells were subjected to well-fed/starved or untreated/treated conditions with and/or without 50 nM BafA1 treatment for the final 2 h and harvested for western blot analysis as described above. SUM149PT cell lines were transfected with 5 nM of siRNA in experiments used to detect the combined effect with Olaparib. In experiments with CASP2 or CASP8 siRNA, cells were harvested at 72 h post-transfection.

### CRISPR-Cas9-based knockout line creation

Single guide RNA sequences targeting the human CASP3 and CASP7 genes were designed using Crispor.org [116] and cloned into PX458 (Cas9-GFP vector with AMP resistance, Addgene, plasmid # 48138) or PX459 (Cas9-Puromycin vector with AMP resistance, Addgene, plasmid # 62988) backbone vector following the protocol provided by Addgene. For gRNA sequences, see S3 Table. Plasmids were then transformed into One Shot Stbl3 chemically competent *Escherichia coli* cells. The resulting CRISPR plasmids were isolated, sequenced and transfected into the SKBR3 or MDA-MB-231 cell line using Lipofectamine 3000 (Invitrogen, L3000-008). After 48 h , GFP-positive individual cells (cells with PX458 vector) were selected by fluorescence-activated cell sorting (FACS) into 96-well plates and cultured to generate monoclonal cell lines. For cells containing CRISPR plasmids in PX459 vector backbone, puromycin-resistant cells were selected by treating with puromycin (Sigma, P9620) at 10 μg/ml concentration and individual clones were obtained by performing serial dilution on the surviving clones. Isogenic knockout clones were validated by western blotting and sequencing.

### CASP, BCL2 and PARP1 construct generation and transfection

For CASP3 and BCL2, a clone from Sino Biological (HG10050-CH) or Addgene (N-FLAG-BCL2, 18003) was obtained, respectively, and used directly for transfection. A PARP1 clone (N-Myc-PARP1, HG11040-NM) was obtained from SinoBiological. CASP7 plasmids containing CASP7-WT, CASP7-inactive (CASP7-C186A), CASP7-ΔPro or non-cleavable at the pro-domain were kindly gifted by Dr. Salvesen [117]. The site-directed mutagenesis method was employed to modify nucleotides in CASP7 or PARP1 as needed. CASP7-p30, CASP7-p29, and CASP7 calpain cleavage sites mutated (CASP7-CCM) constructs were generated by PCR using CASP7-WT as the template. The primers and restriction enzymes used are listed below. Since calpains are known to recognize the three-dimensional structure rather than the sequence of the targeted site of the substrate protein [67,118], and a single amino acid mutation does not prevent the recognition by calpains [40,119], we sequentially altered amino acid residues to generate CASP7-CCM. Multiple amino acid residues at the 1st and 2nd calpain cleavage sites were changed into 'A' until a significant reduction in non-canonical cleavage was obtained (Fig 4C). For the catalytically inactive PARP1 construct (PARP1-CI), three amino acid residues (H862, Y896, and E988) in the catalytic (CAT) domain of PARP1 were sequentially mutated [79]. DNA sequencing was performed to verify the sequence integrity of all the constructs.

To obtain stable cell lines, single or double CASP KO cells (CASP7 KO, CASP3 + 7 DKO) were plated at $2 \times 10^5$ cells/well and transfected with empty vectors (pcDNA3 for CASP7 constructs and pCMV3-C-His for CASP3 constructs) or plasmids containing the CASP constructs, using Lipofectamine 3000 (Invitrogen, L3000-008) as per manufacturer's instructions. CASP3 and CASP7 construct expressing cells were selected with 500 µg/ml Hygromycin and/or 1 mg/ml G418, respectively. Antibodies were used to evaluate the expression of CASP constructs in cells that survived drug treatment. The selection agent(s) was removed for two passages prior to experiments.

The following primers and restriction sites/enzymes were used in making CASP constructs

1. For cloning CASP7-p30 fragment (via Acc65I and XhoI sites)

   Forward – 5′-TGACCTAGGGTACCATGAGTAAGAAGAAGAAAAATGTC-3′

   Reverse – 5′-TGACCTAGCTCGAGCTACTTGTCATCGTCGTCCTTGTA-3′

2. For cloning CASP7-p29 fragment (via Acc65I and XhoI sites)

   Forward – 5′-TGACCTAGGGTACCATGCGATCCATCAAGACCACCCGG-3′

   Reverse – 5′-TGACCTAGCTCGAGCTACTTGTCATCGTCGTCCTTGTA-3′

3. For site-directed mutagenesis of 1st calpain cleavage site for CASP7-CCM construct (FVPS**LFSK**KKK to FVPS**AAAA**KKK)

   5′-/5Phos/CGGTCCTCGTTTGTACCGTCCGCAGCTGCCGCAAAGAAGAAAAAT-3′

4. For site-directed mutagenesis of 2nd calpain cleavage site for CASP7-CCM construct (KNVT**MRS**IKT to KNVT**AAA**IKT)

   5′-/5Phos/AAGAAGAAAAATGTCACCGCTGCCGCAATCAAGACCACCCGGGAC-3′

## Primers used in PARP constructs generation

1. For PARP1-DEVD to DEVA

   Forward – 5′-GGAAAGAGA AAAGGCGATGAGGTGGCGGGAGTGGATGAAGTG-3′

   Reverse – 5′-CACTTCATCCACTCCCGCCACCTCATCGCCTTTTCTCTTTCC-3′

2. For PARP1-DEVD to AEVA

   Forward – 5′-GGAAAGAGAAAAGGCGCAGAGGTGGCGGGAGTGGATGAAGTG-3′

   Reverse – 5′-CACTTCATCCACTCCCGCCACCTCTGCGCCTTTTCTCTTTCC-3′

3. For PARP1-CI (catalytically inactive)

   H862A_Forward – 5′-CGAAGATTGCTGTGGGCAGGGTCCAGGACCACC-3′ H862A_Reverse – 5′-GGTGGTCCTGGACCCTGCCCACAGCAATCTTCG-3′

   Y896A_Forward – 5′-TTTGGTAAAGGGATCGCATTCGCTGACATG GTC-3′

   Y896A_Reverse– 5′-GACCATGTCAGCGAATGCGATCCCTTTACC AAA-3′

   E988A_Forward– 5′-TCTCTACTATATAACGCATACATTGTCTAT GAT-3′

   E988A_Reverse: –5′-ATCATAGACAATGTATGCGTTATATAGTAG AGA-3′

## Crystal violet assay

Cells (parental, CASP3 + 7 DKO, or CASP construct expressing), were plated at $1 \times 10^4$ cells per well in 24-well plates. For CASP3 + 7 knockdown, parental cells were transfected with 10 nM CASP siRNA or Scramble-siRNA at 24 h after seeding. For Olaparib treatment of SUM149PT, cells were transfected with 5 nM of CASP or Scramble-siRNAs. At 48 h after transfection (72 h after seeding), cells were treated with the drug (PI or Olaparib) and allowed to grow in the media containing the drug for 24 h. Then, the cells were washed twice in PBS and continued to grow in drug-free fresh media for another 3 days. For the post-starvation viability assays, at 24 h post-transfection of siRNA, cells were washed in PBS and the media was replaced with fresh media. Cells were then starved in EBSS for 0, 8, 24, 48, and 72 h. After removing the media and washing with PBS, cells were stained with 0.1% crystal violet (Sigma, C6158) for 15 min, washed with distilled water and air dried before taking images using the ChemiDoc MP System (BioRad).

## Immunoprecipitation of CASP7-p30/29 fragments for Edman sequencing

CASP7 or mouse IgG (control) antibody-bound beads were prepared by incubating anti-CASP7 antibody (LSBio C179785), or mouse-IgG (Santa Cruz, Sc2025) with Dynabeads protein G (Thermo Fisher Scientific, 10003D) in IP buffer (Thermo Fisher Scientific, 8159600147) (25 mM Tris–HCl pH 7.5, 150 mM NaCl, 1× PhosSTOP,1% Np-40, 5% Glycerol and 1 mM EDTA with 1× complete mini protease inhibitor (Roche, 4693124001) at 4 °C overnight. CASP7 KO SKBR3 cells or CASP7 KO SKBR3 cells expressing the CASP7-WT construct were grown in T75 flasks for 3 days, fed with fresh growth media or starved for 4 h in EBSS, trypsinized (Gibco, 25300-062), collected and lysed by incubation in cell lysis buffer. Cell lysates were first pre-cleared at 4 °C for 1 h and the lysates were then added to the prepared protein-G beads and mutated at 4 °C for 4 h. The beads with captured proteins were washed 3× with IP wash buffer and elution buffer was added. Captured proteins were released by boiling the beads in 2× SDS sample buffer (117 mM Tris–Cl pH6.8, 4% SDS, 8% glycerol, 0.01% bromophenol blue, 200 mM DTT) at 98 °C for 10 min. The samples were subsequently analyzed by western blotting. After PonceauS staining, the band corresponding to CASP7-p30/29 that contains ∼ 1–2 µg of proteins on PVDF was cut and sent for Edman degradation sequencing (for the first 10 amino acids at N-terminus) at the SPARC BioCentre, Toronto, Canada.

## Fluorescence and phase-contrast microscopy

For staining with DALGreen (Dojindo, NC1879567), seeded cells were preincubated with the dye in the growth media for 30 min at 37 °C, then washed twice with PBS and subjected to autophagy induction by amino acid starvation. Live cells were imaged for fluorescence marker using constant settings (i.e., same laser power and gains). Microscopy images were obtained from a Zeiss Axio Observer (Z1/7) inverted fluorescence microscope equipped with an Apotome.2 and an AxioCam MRm R3 camera (Zeiss). Images were obtained at 20× magnification using the Zen software (version 2.5, blue edition; Zeiss). The number of DALGreen positive punctae in 10 randomly taken images from each cell line were determined using ImageJ64 software and normalized to the number of cells in each image. Total of 500–700 cells were covered and, the experiment was replicated twice. To analyze cell size, phase-contrast images were taken at 20× magnification from 10 randomly selected locations for each cell line (SKBR3 parental and CASP3 + 7 DKO). Each cell outline was manually marked in phase-contrast images, and the area was determined using 'measure' tool in imageJ64. The area measurements were performed in two separate experiments.

## RNA isolation and RT-qPCR analysis

Total RNA was isolated and purified using the RNeasy Plus Mini Kit (QIAGEN, 74104) as per manufacturer's instructions. To remove any contaminating genomic DNA, RNA was treated with DNase 1 (Invitrogen, 18068-015) as per manufacturer's instructions. The quantification of RNA was performed using a Nanodrop Spectrophotometer. The quantitative RT-qPCR was performed to quantitate selected ATG transcript expression levels using One-Step Plus SYBR Green reagent kit (Applied Biosystems, 4385617) in 15 µl reactions containing 100 ng of total RNA and 0.1 µM of each primer on the 7900 sequence detection system (Applied Biosystems). Manufacturer's recommended cycling conditions were used. The ATG proteins and the sequences of forward and reverse primers designed using Primer–BLAST [120] are listed in S2 Table. The mRNA expression levels were determined by the Comparative Cycle Threshold method ($2^{-[delta][delta]Ct}$) normalizing to internal reference gene ($\beta$actin) . The expression is presented as the fold change relative to the control. RT-qPCR analyses were done in two or three technical replicates and experiments were repeated at least thrice.

## DNA fragmentation/Comet assay

The level of DNA damage (single and double strand breaks) was evaluated by the comet assay (single cell-gel electrophoresis) as previously described [121]. Briefly, cells were washed in PBS (2-3×), trypsinised, harvested in cold PBS and the supernatant was removed after spinning for 5 min at 1,200 rpm at 4 °C. Each cell pellet was resuspended in cold PBS and gently mixed with 1% low melting agarose (Invitrogen, 16500-100) solution at a ratio of 1:10 (v/v). A drop from each cell-agarose mixture was immediately placed on a 1% agarose pre-coated glass microscopic slide, covered with a cover slip, and placed flat for 30 min at 4 °C. Next, the coverslips were removed gently, and the slide was immediately placed in 4 °C lysis solution (2.5 M NaCl, 100 mM disodium EDTA, 10 mM Tris base, and 200 mM NaOH) overnight at 4 °C. Next, slide was incubated in alkaline unwinding buffer (pH > 13) (200 mM NaOH, 1 mM EDTA) for 20 min at RT in the dark before being subjected to gel electrophoresis in alkaline buffer (200 mM NaOH, 1 mM EDTA) at ~ 20 V (300 mA) for 30 min at 4 °C. Slides were rinsed by immersing in $dH_2O$ for 5 min (2×), DNA was fixed by immersing in 70% ethanol for 5 min, and dried at 37 °C for 10 min before stained with GelRed (Biotium, 41,003) (1/3,000 dilution) for 30 min at RT in darkness. The slide was rinsed in water, dried, and mounted with slow-fade mounting media. Images were taken using a fluorescent microscope at 10× objective and DNA damage was quantified as the comet tail moment (product of the tail length × the fraction of total DNA in the tail) using CometScore 2.0 software [122]. Approximately, 200 randomly selected nucleoids were scored per each treatment. Experiments were repeated twice.

## In silico co-essentiality and genetic interaction network mapping

*In silico* genetic interaction screening was performed using GRETTA (v0.99.2) [95] on the R statistical software (v4.2.2) [123]. We computed on the cancer DepMap public release version 22Q2 [124] data from FigShare https://figshare.com/articles/dataset/DepMap_22Q2_Public/19700056/2; accessed on 11 August 2022) according to Takemon and Marra (2023). GRETTA was used to identify cell lines with combined low or high CASP3 and CASP7 protein expression. We identified a group of pan-cancer cell lines below the 10th percentile of CASP3 and CASP7 protein expression (16 cell lines; low expressors) and those above the 90th percentile of both CASP3 and CASP7 protein expression (10 cell lines; high expressors) as the control comparator. The normalized (scaled and mean centered) gene and protein expression values for the selected cell lines were extracted from Nusinow and colleagues (2020) [98] using GRETTA for comparison between the low and high expressor groups.

Genetic interactors of combined CASP3 and CASP7 low expression were predicted using GRETTA, which performed pairwise Mann–Whitney $U$-tests between the high expressors and low expressors cancer cell line groups for all 17,386 genes to obtain $p$-values. $P$-values were adjusted for multiple testing using a permutation (10,000 randomized resampling). Candidate genetic interactors of combined low CASP3 and CASP7 expression were called using a threshold of adjusted Mann–Whitney $U$-test $p$-value < 0.05 and a median lethality probability > 0.5 in at least one group as previously defined [95]. To highlight candidate lethal interactors, genetic interaction scores computed by GRETTA were used to generate lethal genetic interaction scores by collapsing genetic interaction scores below zero to zero. A higher interaction score indicates an increased likelihood that a gene knockout is lethal in the test group. Rank is based on lethal GI scores.

## Enrichment analysis and gene function annotation

We used clusterProfiler (v4.6.0) [125] to annotate GO biological processes of genes that were candidate co-essential or genetic interactors (unadjusted $p$-value < 0.05). Given many related GO terms with similar sets of genes associated with them, we calculated the degree of overlapping genes associated between GO terms using the Jaccard indices and performed hierarchical clustering to summarize their roles, as performed by Takemon and colleagues [95]. The number of distinct clusters was determined using gap statistics, which calculated the optimal number of clusters (up to 20 clusters) by iteratively bootstrapping 10,000 times.

## Statistical analysis

In each graph, error bars represent standard error of at least $n = 3$ independent experiments. As indicated in the legends, statistical significance was calculated by Student's $t$-tests (for comparisons between two samples) or analysis of variance (ANOVA) with appropriate post-tests for multiple comparisons (Tukey's, Dunnett's, or Sidak's) using GraphPad Prism version 8.4.1 for Windows (GraphPad Software, San Diego, California, USA, www.graphpad.com). Significance ($p$-values) indicated are relative to the control unless otherwise indicated. $P$-values less than 0.05% were determined as significant.

Significance is indicated as follows unless otherwise noted: $*P < 0.05$, $**P < 0.01$, $***P < 0.001$, $****P < 0.0001$, NS, not significant.

## Supplementary information

**S1 Fig. Loss of CASP3 and CASP7 suppresses starvation-induced autophagy in MDA-MB-231 cells; related to Fig 1.** (A, B) Representative western blots showing levels of CASP3 or CASP7 following treatment with scramble-siRNA, CASP3 siRNA or CASP7 siRNAs used in experiments. (**C**) Representative western blots showing levels of CASP3 and CASP7 in knockout lines of CASP3 and/or CASP7 in SKBR3 cells, cultured in fed conditions (F; fresh DMEM) or subjected to amino acid starvation (S) in EBSS for 8 h. (**D**) Representative western blots of indicated proteins from CASP3 + 7 DKO SKBR3 cells cultured in fed conditions (fresh DMEM) or subjected to amino acid starvation in EBSS for 24 h, in the absence or presence of BafA1(50 nM) for the final 2 h. (**E**) Quantification of LC3B-based autophagy flux shown in (D). The levels of LC3BII were normalized to loading control and shown relative to the fed (fresh DMEM) in the absence of BafA1. (**F**) Representative western blots of indicated proteins from SKBR3 parental and multiple CASP3 + 7 DKO cell lines (CRISPR-Cas9 mediated), cultured in fed conditions (fresh DMEM) or subjected to amino acid starvation in EBSS for 8 h. BafA1 (50 nM) was added for the final 2 h of culture. (**G**) Representative western blots of indicated proteins from MDA-MB-231 cells cultured in fed conditions (fresh DMEM) or subjected to amino acid starvation in EBSS for various time periods, in the absence (top)

or presence (bottom) of BafA1(50 nM) for the final 2 h. (**H**) Quantification of LC3B-based autophagy flux shown in (G). The levels of LC3BII (in the presence of BafA1) were normalized to loading control and shown relative to the fed (fresh DMEM) control. (**I**) Representative western blots of indicated proteins from MDA-MB-231 cells transfected with scramble, CASP3 and/or CASP7 siRNAs (48 h) and then continued to be cultured in fed conditions (fresh DMEM) or starved in EBSS for 8 h. BafA1 (50 nM) was added for the final 2 h. (**J**) Quantification of LC3BII-based autophagy flux shown in (I). The levels of LC3BII in starved cells were normalized to loading control and shown relative to the starved scramble-siRNA control. (**K**) Representative western blots of indicated proteins from MDA-MB-231 parental and CASP3 + 7 DKO cells cultured in fed conditions (fresh DMEM) or starved in EBSS for 8 or 24 h. BafA1 (50 nM) was added for the final 2 h. (**L**) Quantification of LC3BII-based autophagy flux shown in (K). The levels of LC3BII in starved cells were normalized to loading control and shown relative to the parental control (all from starved conditions in the presence of BafA1). (**M**) Representative phase contrast microscopy images of SKBR3 parental and CASP3 + 7 DKO cells starved in EBSS for 8 h. Scale bars, 50 μm. (**N**) Quantification of cell size (area) in cells shown in (M). $n = 2$, in each replicate, cells in 10 random images were measured. In graphs, all data are shown as mean ± SEM. $n = 3$ independent experiments. *$P < 0.05$, **$P < 0.01$, ***$P < 0.001$, ****$P < 0.0001$, NS, not significant. In H, J, and L, one-way ANOVA with Dunnett's post-test. In N, one-way ANOVA with Tukey's post-test. The numerical data presented in this figure can be found in S2 Data.
(TIF)

**S2 Fig. Loss of CASP3 and CASP7 suppresses compensatory autophagy induced by MG132 or Bortezomib in MDA-MB-231 and SKBR3 cells, sensitizing cells to proteasome inhibitors; related to Fig 2.** (A) Representative western blots of indicated proteins from MDA-MB-231 cells treated with proteasome inhibitor (MG132) at increasing dosage for 24 h in the presence (top) or absence (bottom) of BafA1(50 nM) for the final 2 h. (**B**) Quantification of LC3B-based autophagy flux in MDA-MB-231 shown in (A). The levels of LC3BII were normalized to loading control and shown relative to the untreated control (NT). (**C**) Graph showing proteasome activity in MDA-MB-231 cells in response to increasing concentrations of MG132 as depicted by chymotrypsin-like, caspase-like, and trypsin-like activity measured by the Proteasome glow assay kit. (**D**) Representative western blots of indicated proteins from MDA-MB-231 cells transfected with scramble, CASP3 and/or CASP7 siRNAs (48 h) and continued to be cultured in vehicle DMSO or MG132 (0.5 μM) treated fresh DMEM for 24 h. BafA1(50 nM) was added for the final 2 h. (**E**) Quantification of LC3B-based autophagy flux in proteasome inhibitor (MG132) treated MDA-MB-231 cells shown in (D). The levels of LC3BII in MG132-treated cells were normalized to loading control and shown relative to the MG132-treated scramble-siRNA control. (**F**) Representative western blots of indicated proteins from SKBR3 cells transfected with scramble, CASP3 and/or CASP7 siRNAs (48 h) and continued to be cultured in vehicle DMSO or Bortezomib- (1 nM) treated fresh DMEM for 24 h. BafA1 (50 nM) was added for the final 2 h. (**G**) Quantification of LC3B-based autophagy flux in proteasome inhibitor- (Bortezomib) treated SKBR3 cells shown in (F). The levels of LC3BII in Bortezomib-treated cells were normalized to loading control and shown relative to the Bortezomib-treated scramble-siRNA control. (**H–M**) Representative images of crystal violet assay plates and quantification of percentage of cell viability. The indicated siRNA transfected cells were grown for 2 days or the Parental and DKO cells were grown in normal media for 2 days and treated with indicated concentrations of proteasome inhibitors (MG132 or Bortezomib) for 24 h and were then continued to be cultured in drug free media for another 3 days before subjected to crystal violet assay. Graphs show the percentage of stained (viable)

cells at each concentration normalized to respective untreated cells. MDA-MB-231 cells transfected with scramble or CASP3 + 7siRNA and treated with MG132 (H), and its quantification is in (I). Bortezomib-treated SKBR3 parental and CASP3 + 7 DKO cells (J) and its quantification (K). MDA-MB-231 cells transfected with scramble or CASP3,7 siRNAs and treated with Bortezomib (L) and its quantification (M). (**N–O**) Representative images of crystal violet assay plates and quantification of percentage of cell viability. The indicated siRNA transfected cells were grown for 1 day, replaced with fresh media or subjected to starvation for indicated duration before subjected to crystal violet assay. Graphs show the percentage of stained (viable) cells at each starvation time point normalized to fed cells. In graphs, all data are shown as mean ± SEM. $n$ = 3 independent experiments. *$P < 0.05$, **$P < 0.01$, ***$P < 0.001$, ****$P < 0.0001$, NS, not significant. In B, C, E, and G, one-way ANOVA with Dunnett's post-test. In I, M, and O, two-way ANOVA with Sidak's post-test. In K mixed model with Sidak's post-test. The numerical data presented in this figure can be found in S2 Data.
(TIF)

**S3 Fig. Rapamycin-induced autophagy does not depend on CASP3 and CASP7; related to Fig 3.** (A, B) Representative western blots of indicated proteins from SKBR3 (A) or MDA-MB-231 (B) parental and CASP3 + 7 DKO cells untreated or treated with mTOR inhibitor rapamycin (Rap) 10 nM for 24 h in the absence or presence of BafA1 (50 nM) for the final 2 h. The levels of phosphorylated and unphosphorylated p70S6K were used as markers for mTOR activity. (**C**) Representative western blots showing the effects of rapamycin on the formation of CASP7-p29/p30 bands in MDA-MB-231 cells. Cells grown in normal culture media for 4 days were continued to be cultured in fed (fresh DMEM) conditions, starved in EBSS for 4 h, treated with 10 nM rapamycin (Rap) for 24 h or subjected to both starvation and Rap treatment for 24 h. The p70S6K and p70S6K-PO4 immunolabeling was used as an mTOR activity reporter. In each $n = 2$ independent experiments.
(TIF)

**S4 Fig. During non-lethal stress, non-canonical cleavage of CASP7 occurs at calpain cleavage sites; related to Fig 4.** (**A**) Representative western blot showing CASP7 immunolabeling in CASP3 + 7 DKO SKBR3 cells stably expressing CASP7-WT, CASP7 ΔPro, and CASP7 prodomain cleavage blocked constructs cultured in fed conditions (fresh DMEM) or starved in EBSS for 4 h. Arrowhead indicates CASP7-ΔPro fragment (33.5 kDa) and arrow indicates CASP7-p29/30 fragments. $n = 1$. (**B**) Representative western blot showing immunoprecipitation of CASP7-p29/30 fragment(s) for Edman sequencing. CASP7 KO SKBR3 cells stably expressing CASP7-WT construct were cultured in fed conditions (fresh DMEM) or starved in EBSS for 2 h prior to immunoprecipitation with CASP7 antibody. Lysate are from CASP7-WT expressing and CASP7-KO cells. IP with anti-mouse CASP7 and anti-mouse IgG antibodies are shown. $n = 2$ independent experiments. (**C**) A schematic showing mouse CASP7 with predicted calpain cleavage sites identified by the Deepcalpain algorithm. (**D**) Representative western blots of indicated proteins from CASP3 KO SKBR3 cells transfected with scramble, calpain 1, or calpain 2 siRNAs (48 h) and then continued to be cultured in vehicle DMSO or MG132 (0.5 μM) in fresh DMEM for 24 h and with BafA1(50 nM) in the final 2 h. (**E**) Quantification of LC3B-based autophagy flux in MG132-treated cells shown in (D) relative to MG132-treated scramble-siRNA control. (**F**) Schematic showing the amino acid sequence between two non-canonical cleavage sites in CASP7. PARP binding site is shown. In graph (E), data are shown as mean ± SEM. $n$ = 3 independent experiments. *$P < 0.05$, **$P < 0.01$, ***$P < 0.001$, ****$P < 0.0001$, NS, not significant, with one-way ANOVA with Dunnett's post-test. The numerical data presented in this figure can be found in S2 Data.
(TIF)

**S5 Fig. Non-canonical CASP7 cleavage sites flank a PARP1 binding site; related to Fig 5.** (A–E) Reverse transcription polymerase chain reaction (RT-qPCR) analyses of key ATG gene expression in SKBR3 parental or CASP3 + 7 DKO cells under well-fed conditions. (**F**) RT-qPCR analyses of ATG7 in SKBR3 parental, CASP3 + 7 DKO, or CASP3 + 7 DKO cells re-expressing CASP3 + 7-WT constructs, treated with vehicle DMSO (NT) or 0.5 μM of MG132 for 24 h. (**G**) Representative western blots of indicated proteins from SKBR3 parental or CASP3 + 7 DKO cells, transiently transfected with indicated Myc-tagged PARP constructs (PARP1-WT, PARP1-DEVA, PARP1-AEVA, or catalytically inactive PARP1-CI) or vector control (v) for 48 h. endo = endogenous PARP1. (**H-I**) Quantification of cleaved-PARP1 (H) and PAR levels (I) shown in (G). (**J**) RT-qPCR analyses of LC3B transcripts in basal conditions in SKBR3 parental or CASP3 + 7 DKO cells transiently transfected with indicated Myc-tagged PARP constructs (PARP1-WT, PARP1-DEVA, PARP1-AEVA, or PARP1-CI) or vector control (v). Data of one replicate with relatively high LC3B levels is circled in red. (**K**) Representative western blots of indicated proteins from SKBR3 parental or CASP3 + 7 DKO cells, transiently transfected with indicated Myc-tagged PARP constructs (PARP1-WT, PARP1-DEVA, PARP1-AEVA, or PARP1-CI) or vector control (v) for 48 h and with BafA1 (50 nM) in the final 2 h. endo = endogenous PARP1 (**L**) Quantification of LC3B-based autophagic flux in basal conditions, shown in (K). (**M-P**) Representative western blots of indicated proteins from SKBR3 cells transfected with scramble, calpain 1, calpain 2, CASP6, cathepsin B (cathB), cathepsin D (cathD) or CASP8 siRNAs (48 h) and treated with vehicle DMSO or MG132 (0.5 μM) for 24 h (M and N) or grown in untreated media (in O and P). Each with at least n = 2 independent experiments. In graphs, unless otherwise noted, all data are shown as mean ± SEM. n = 3 or more independent experiments. *$P < 0.05$, **$P < 0.01$, ***$P < 0.001$, ****$P < 0.0001$, NS, not significant. A-E with Student $T$ test (unpaired). In F, H, I, J, and L, one-way ANOVA with Dunnett's post-test. The numerical data presented in this figure can be found in S2 Data.
(TIF)

**S6 Fig. Cell viability is unaffected in CASP3 + 7 DKO cells expressing CASP7-CCM, p30 or p29; related to Fig 6.** (A) Representative western blot of CASP7 immunolabeling from SKBR3 parental or CASP3 + 7 DKO cells stably expressing CASP7-CCM, CASP7-p30 or CASP7-p29 constructs, treated with vehicle DMSO or proteasome inhibitor MG132 (0.5 μM) for 24 h. n = 2 independent experiments. (**B**) Representative images of crystal violet assay plates and quantification in cells expressing CASP7-CCM, CASP7-p30 or CASP7-p29 constructs. Cells were grown in standard media for 2 days and treated with indicated concentrations of proteasome inhibitor Bortezomib for 24 h and continued to grow in drug free media for another 3 days before subjected to crystal violet assay. (**C**) Quantification of cell viability data presented in (B). The percentage of stained (viable) cells at each concentration was normalized to respective untreated cells. Data are shown as mean ± SEM. n = 3 independent experiments. NS, not significant. Two-way ANOVA with Tukey's post-test. (D) Graphical representation of current working model. The numerical data presented in this figure can be found in S2 Data.
(TIF)

**S7 Fig. *in silico* analyses identify CASP3 and CASP7 genetic interactions; related to Fig 7.** (A) Low and high levels of CASP3 and CASP7 expressing cancer cell lines identified and analyzed in Fig 7H and I. (**B**) Heatmap showing Jaccard index similarities between GO terms. Based on annotated GO biological processes of genes that were candidate co-essential or genetic interactors (unadjusted p-value < 0.05). GO terms were summarized into 16 distinct clusters using Jaccard index-based hierarchical clustering. The numerical data presented in this figure can be found in S2 Data. The code related to S7A and S7B Fig is publicly available in a GitHub repository

(https://github.com/MarraLab/Caspase_GRETTA_analysis) and archived on Zenodo (https://doi.org/10.5281/zenodo.14722298)
(TIF)

**S1 Raw Images. Original western blots used for** Figs 1A, 1C, 1E, 1G, 2A, 2D, 2F, 2H, 3A-J, 4A, 4C-E, 5A-B, 5E-F, 5J, 5N, 6A, 6E, 6G, 6M, 6O, 7C-D, **S1A-D, S1F-G, S1I, S1K, S2A, S2D, S2F, S3A-C, S4A-B, S4D, S5G, S5K, S5M-P and S6A** . The green box indicates the area of the blot that was used for the figure when only a portion of the blot was used. Lanes have not been rearranged; the order of the lanes is same as the order shown in figures.
(PDF)

**S1 Data. The underlying numerical data in** Figs 1B, 1D, 1F, 1H, 1J, 2B-C, 2E, 2G, 2I, 2K, 4F-I, 5C-D, 5G-H, 5I, 5K-M, 5O, 6B, 6D, 6F, 6H, 6J, 6L, 6N, 6P, 7B, 7E **and** 7G-I.
(XLS)

**S2 Data. The underlying numerical data in S1E, S1H, S1J, S1L, S1N, S2B-C, S2E, S2G, S2I, S2K, S2M, S2O, S4E, S5A-F, S5H-J, S5L, S6C and S7B Figs.**
(XLSX)

**S1 Table. siRNAs used in knockdown experiments.**
(DOCX)

**S2 Table. Forward and Reverse primer sequences used for qRT-PCR.**
(DOCX)

**S3 Table. Key resources table.**
(DOCX)

## Acknowledgements

The authors would like to thank Arlene Gidda for valuable comments on the manuscript, Gorski lab members for helpful discussions, Dr. Peter Eirew for assistance with providing the PDX samples, Dr. Nancy dos Santos and Nicole Wretham for murine tissues and Dr. Guy S. Salvesen for FLAG-tagged CASP7 constructs, Susanna Y. Chan for technical advice for Comet assay, and Dr. Stephanie McInnis for project management support. Graphical representation of current working model was created using BioRender.com.

## Author contributions

**Conceptualization:** Gayathri Samarasekera, Nancy E. Go, Courtney Choutka, Suganthi Chittaranjan, Sharon M. Gorski.

**Data curation:** Gayathri Samarasekera, Nancy E. Go, Courtney Choutka, Jing Xu.

**Formal analysis:** Gayathri Samarasekera, Nancy E. Go, Yuka Takemon, Jennifer Chan, Suganthi Chittaranjan.

**Funding acquisition:** Gregg B. Morin, Sharon M. Gorski.

**Investigation:** Gayathri Samarasekera, Nancy E. Go, Courtney Choutka, Jing Xu, Yuka Takemon, Jennifer Chan, Michelle Chan, Shivani Perera, Suganthi Chittaranjan.

**Methodology:** Gayathri Samarasekera, Nancy E. Go, Courtney Choutka.

**Project administration:** Sharon M. Gorski.

**Resources:** Samuel Aparicio, Sharon M. Gorski.

**Supervision:** Marco A. Marra, Sharon M. Gorski.

**Validation:** Gayathri Samarasekera, Nancy E. Go.

**Visualization:** Gayathri Samarasekera, Nancy E. Go, Yuka Takemon, Suganthi Chittaranjan, Sharon M. Gorski.

**Writing – original draft:** Gayathri Samarasekera, Sharon M. Gorski.

**Writing – review & editing:** Gayathri Samarasekera, Nancy E. Go, Courtney Choutka, Jing Xu, Yuka Takemon, Jennifer Chan, Michelle Chan, Shivani Perera, Samuel Aparicio, Gregg B. Morin, Marco A. Marra, Suganthi Chittaranjan, Sharon M. Gorski.

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
