## [Editor Report · Decision Letter 0]

5 Mar 2024

Dear Dr Gorski, 

Thank you for submitting your manuscript entitled "Caspase 3 and caspase 7 promote adaptation to non-lethal stress through PARP1" for consideration as a Research Article by PLOS Biology. Please accept my apologies for the delay in getting back to you with feedback.

Your manuscript has now been evaluated by the PLOS Biology editorial staff, as well as by an academic editor with relevant expertise, and I am writing to let you know that we would like to send your submission out for external peer review.

Please note, however, that the outcome of our discussion of your manuscript with our academic editor is that we have some reservations as to the depth of analysis and the overall strength of the mechanistic insights and physiological relevance offered by your data. We would need to be persuaded by the reviewers that the paper has the potential after revision to offer the significant strength of advance that we require for publication in order to pursue it further for PLOS Biology.

Once your full submission is complete, your paper will undergo a series of checks in preparation for peer review. After your manuscript has passed the checks it will be sent out for review. To provide the metadata for your submission, please Login to Editorial Manager (https://www.editorialmanager.com/pbiology) within two working days, i.e. by Mar 07 2024 11:59PM.

Kind regards,

Richard

Richard Hodge, PhD

rhodge@plos.org

PLOS

---

## [Decision Letter · Decision Letter 1]

5 Apr 2024

Dear Dr Gorski,

Thank you for your patience while your manuscript "Caspase 3 and caspase 7 promote adaptation to non-lethal stress through PARP1" was peer-reviewed at PLOS Biology. Please accept my apologies for the delays that you have experienced during the peer review process. Your manuscript has now been evaluated by the PLOS Biology editors, an Academic Editor with relevant expertise, and by three independent reviewers. 

In light of the reviews, which you will find at the end of this email, we would like to invite you to revise the work to thoroughly address the reviewers' reports.

As you will see below, the reviewers are generally positive and think that your study is interesting and well done. However, the reviewers raise overlapping concerns about the over-interpretation of some of the results and the level of mechanistic insight into how caspase-3/7 prevents PARP cleavage. Both Reviewers #1 and #3 suggest an experiment using a PARP cleavage-resistant mutant to see if it can rescue the caspase3/7 knockout phenotype. In addition, Reviewer #2 raises concerns that it is not clear how Caspase-3/7 are being activated in the model and notes that it is important to investigate the potential role of the MOMP pathway.

Given the extent of revision needed, we cannot make a decision about publication until we have seen the revised manuscript and your response to the reviewers' comments. Your revised manuscript is likely to be sent for further evaluation by all or a subset of the reviewers.

**IMPORTANT - SUBMITTING YOUR REVISION**

*Re-submission Checklist*

*Published Peer Review*

*PLOS Data Policy*

*Blot and Gel Data Policy*

Sincerely,

Richard

Richard Hodge, PhD

rhodge@plos.org

REVIEWS:

Reviewer #1: Gayathri Samarasekera and colleagues have submitted a manuscript titled "Caspase 3 and caspase 7 promote adaptation to non-lethal stress through PARP1" for review. The authors propose that unconventional and non-apoptotic functions of Caspase-3 and -7 are essential for stress adaptation, mediated by Parp1 modulation affecting autophagy and the DNA damage response. While some sections present convincing data, such as alternative cleavage of Casp-3 and -7 under non-lethal stress, and dependency on calpains for this cleavage, other results are occasionally overinterpreted and confusing. The rationale for experimental design and result descriptions is unclear in several sections, and it is uncertain whether the effects described are specific to tumour cells or have broader biological significance. Additional experimental work and clearer organisation and description of the text are needed prior publication. Specific points for the authors to address are outlined below.

1. The authors should consider demonstrating the autophagy effects 24 hours post-starvation in caspase-deprived cells without Bafilomycin treatment, even if the resolution of the experiment is less clear.

2. The conclusions drawn from Figure 1I-J are problematic because the number of Dalgreen dots per cell, used as a proxy for autophagy, is not normalized for cell size. Given the significantly reduced size of double Casp-KO cells, it is unclear how the authors conclude that there is less autophagy in absolute terms.

3. Regarding Figure 4C, the rationale for creating Casp7 calpain-resistant mutants using multiple amino acid substitutions instead of a single S>A conversion is unclear. This approach could unpredictably alter the general folding of the caspase template. Additionally, why the generation of the -29/30 fragments is not fully abolished in these mutants (as shown in the Western blot in Figure 4C) should be addressed.

4. Starting from Figure 5, the text becomes unclear. The authors suggest that the autophagy effects are somehow related to Parp1. How Parp1 is cleaved in the absence of Casp-3 and -7 needs to be elucidated.

5. The authors mention that excess Parp1 cleavage compromises Parp-1 activity. Is a catalytically dead version of Parp1 sufficient to prevent the autophagy effects observed in Casp-3-7 KO cells?

6. According to the authors' results, a non-cleavable caspase version of Parp1 would be expected to sustain Poly(ADP-ribosyl)ation and autophagy gene transcription. Is this prediction accurate?

7. Casp-3 and Casp-7 appear to act redundantly regarding Parp-1, but the specificity mechanisms for the effects on γ-H2AX are specific to Casp-7. The biological relevance of this phenotype and why only double deficiency in Casp3 and Cas-7 can mimic the effects of Parp1 deprivation should be discussed. Additionally, it is intriguing that, as depicted in the model Figure, both Caspase-3 and -7 are required for all the phenotypes, even though Parp-1 cleavage is mainly associated with Caspase-7. Is there an additive effect in the absence of Caspase-3 when eliminating Caspase-7 activity?

Reviewer #2: In this manuscript by Samarasekera et al, the authors present a non-apoptotic function for caspases-3 and -7 in promoting stress induced cryoprotective autophagy. Overall this is an interesting paper and the concept of non-canonical cleavage of caspase-7 inducing this in the absence of cell death is compelling. The experiments are well designed and the data presented is convincing. I only have a couple of suggestions

1. It is not clear how caspase-3 and -7 are being activated in this context. Is this through the mitochondrial pathway? It has been shown that GAPDH can promote autophagy and survival in cells that have undergone mitochondrial outer membrane permeabilization (MOMP) in the absence of caspase activity. While I don't think this is the same pathway, it would be good to rule in or out any effect of mitochondrial permeabilization in this context. Overexpression of Bcl2 or BclxL would be sufficient to address this to see if it mimics the caspase-7/3 knockout. 

2. In the calpain knockdown experiments in Figure 4 the results (F-I) do not appear to be statistically significant. Do these cells still express caspase-3? If this was done in a caspase-3 knockout/knockdown background the differences on LC3BII levels might be more extensive (assuming that caspase-3 is processed by a different mechanism) 

Reviewer #3: Caspase 3 and caspase 7 promote adaptation to non-lethal stress through PARP1

Samarasekera et al.

General comments

This is an interesting series of experiments that are well presented and clearly explained.

The authors make four main claims in the manuscript: (1) CASP3 and CASP7 promote cytoprotective autophagy induction in response to nonlethal stress; (2) CASP7 processing upon non-lethal stress is distinct from its processing in apoptosis; (3) PARP cleavage is increased and PARP1 activity and DNA Damage Response (DDR) signaling are reduced following CASP3 and CASP7 inhibition; and, (4) loss of CASP3 and CASP7 is synthetic lethal with loss of BRCA1.

Claim 1 is in two parts: cytoprotection and autophagy. Autophagy is studied with a range of assays for different endpoints and is convincing. The claim for cytoprotection relies on Figure 2J and 2K and a supplementary figure which show the effect of MG132 and Bortezomib in other cell types which show decreased viability in Casp7/7 DKO cells compared to wild-type cells. The data presented are clear, but there are no data showing cytoprotection for starvation. Are Casp3/7 DKO cells more vulnerable than wild type cells to starvation?

Claim 2 is supported by the data presented using a combination of approaches including rescue experiments in DKO cells. The authors link the non-cannonical cleavage to a previous report that calpain can cleave and activate caspase-7.

Claim 3 is supported by assays for PARP cleavage, PARylation and Histone 2X phosphorylation. The data are clear.

Claim 4 is supported by data comparing human tumor cell lines that have or lack BRAC1.

Overall the experiments are clearly described, the data are well presented and the authors' conclusions are clear. However, although the Casp3/7 DKO affects PARP, and inhibiting PARP or knocking out Casp3/7 sensitizes cells, how caspase-3/7 prevents PARP cleavage is unaddressed.

Comments on the Introduction

The authors summarize the evidence that caspases interact with components of autophagy in both mammalian and fly models, which includes both initiator and effector caspases. In the light of this summary, the meaning of the final sentence of the paragraph: "whether effector caspases have an evolutionarily conserved function is unknown" is not clear. I suggest emphasizing that mammalian initiators were known to interact with autophagy proteins, and that the current study investigated effectors.

Comments on the experiments and the results section

The authors show evidence that calpain cleavage of caspase-7 and that the calpain cleavage is near PARP binding exosite in caspase-7. However the p30 form (with exosite) and p29 (lacking exosite) have similar functions and the exosite may therefore not be relevant to the non-apoptotic effects of caspase-7. The authors also show that loss of caspase-7 increases PARP cleavage and decreases PARP activity. In addition, they link PARP activity to the expression of a couple of autophagy genes. The authors link these observations to cancer by showing synthetic lethality between BRCA1 and the loss of caspase. 

There is no further explanation of the mechanism of (1) the PARP cleavage seen in the DKO cells or (2) how caspase-7 suppresses PARP cleavage. The authors show that loss of caspase affects PARP and loss of caspases mimics PARP inhibition and conclude that caspases "promote adaptation through PARP" (my underlining). The authors' claim is very strong and I agree that it seems likely that caspase act through PARP, but the authors didn't test this idea. For example, does overexpression of a PARP mutant that cannot be cleaved rescue the caspase DKO phenotype?

Comments on the Discussion 

I think the authors could be clearer about their position, re-writing some sections or by providing experimental evidence if they have it. 

1. Use of "differentially" is unnecessary in "CASP7 is differentially cleaved, resulting in stable fragments (CASP7-p29/30) that are distinct from canonical cleaved-CASP7 fragments (CASP7-p20/p12) reported in apoptosis."

2. The statement: "Overall, this study shifts the current paradigm of apoptotic caspase-PARP1 interactions to one that involves non-canonical caspase cleavage and the promotion of cell adaptation and survival pathways at the onset of cellular stress or in non-lethal cellular stress conditions" is a key claim of the authors. I take interactions to mean binding, but do the authors mean that PARP binds Caspase-7 activated by non-canonical cleavage? Later in the text the authors write:

"Since CASP7-p29 does not have an intact exosite, one would expect (my underlining) it to differently modulate PARP1 and downstream stress response pathways. Despite this key difference, expression of either exogenous p29 or p30 alone were able to enhance the autophagy response or the DNA damage response (γ-H2AX), revealing a potential functional similarity between these non-canonical CASP7 fragments." As one fragment that might bind PARP and one that cannot exert similar effects isn't this evidence that the effects of the non-canonical fragments are independent of PARP binding? 

3. Other specific question to be more clearly addressed include:

Non-canonical cleavage of caspase-7 is reported to increase the activity of caspase-7, how does increased activity fit with a non-apoptotic outcome?

Comments on the Figures and Figure legends

Typographical error Figure 6M. Two periods at the sentence's end.

Summary

This is a very interesting manuscript, with well performed experiments that are clearly presented. It is very well suited to PLOS Biology. The major point the authors should address experimentally is whether an "uncleavable" PARP can rescue the DKO phenotype. This opinion is the basis of the decision "Major Revision". The other points can be addressed by revising the indicated areas of text.

---

## [Decision Letter · Decision Letter 2]

18 Dec 2024

Dear Dr Gorski,

Thank you for your patience while we considered your revised manuscript "Caspase 3 and caspase 7 promote adaptation to non-lethal stress and their dual loss phenocopies PARP1 inhibition" for publication as a Research Article at PLOS Biology. This revised version of your manuscript has been evaluated by the PLOS Biology editors, the Academic Editor and the original reviewers.

Based on the reviews, I am pleased to say that we are likely to accept this manuscript for publication, provided you satisfactorily address the following data and other policy-related requests that I have provided below (A-G). Please also ensure that the figure labels are correctly cited in the text as requested by Reviewer #2.

(A) We routinely suggest changes to titles to ensure maximum accessibility for a broad, non-specialist readership. In this case, we would suggest a minor edit to the title, as follows. Please ensure you change both the manuscript file and the online submission system, as they need to match for final acceptance:

“Caspase-3 and caspase-7 modulate promote cytoprotective autophagy during non-lethal stress conditions in human breast cancer cells"

(B) In the animal ethics statement in the Methods section, please provide the specific approval number provided by the IACUC to conduct the study.

(C) You may be aware of the PLOS Data Policy, which requires that all data be made available without restriction: http://journals.plos.org/plosbiology/s/data-availability. For more information, please also see this editorial: http://dx.doi.org/10.1371/journal.pbio.1001797

-Supplementary files (e.g., excel). Please ensure that all data files are uploaded as 'Supporting Information' and are invariably referred to (in the manuscript, figure legends, and the Description field when uploading your files) using the following format verbatim: S1 Data, S2 Data, etc. Multiple panels of a single or even several figures can be included as multiple sheets in one excel file that is saved using exactly the following convention: S1_Data.xlsx (using an underscore).

-Deposition in a publicly available repository. Please also provide the accession code or a reviewer link so that we may view your data before publication. 

Figure 1B, 1D, 1F, 1H, 1J, 2B-C, 2E, 2G, 2I, 2K, 4F-I, 5C-D, 5G-H, 5I, 5K-M, 5O, 6B, 6D, 6F, 6H, 6J, 6L, 6N, 6P, 7B, 7E, 7G-I, S1E, S1H, S1J, S1L, S1N, S2B-C, S2E, S2G, S2I, S2K, S2M, S2O, S4E, S5A-F, S5H-J, S5L, S6C, S7B

(D) Please also ensure that each of the relevant figure legends in your manuscript include information on *WHERE THE UNDERLYING DATA CAN BE FOUND*, and ensure your supplemental data file/s has a legend.

(E) We require the original, uncropped and minimally adjusted images supporting all blot and gel results reported in the following Figures:

Figure 1A, 1C, 1E, 1G, 2A, 2D, 2F, 2H, 3A-J, 4A, 4C-E, 5A-B, 5E-F, 5J, 5N, 6A, 6E, 6G, 6M, 6O, 7C-D, S1A-D, S1F-G, S1I, S1K, S2A, S2D, S2F, S3A-C, S4A-B, S4D, S5G, S5K, S5M-P, S6A

We will require these files before a manuscript can be accepted so please prepare and upload them now. Please carefully read our guidelines for how to prepare and upload this data: https://journals.plos.org/plosbiology/s/figures#loc-blot-and-gel-reporting-requirements

(F) Please ensure that your Data Statement in the submission system accurately describes where your data can be found and is in final format, as it will be published as written there. 

(G) Per journal policy, if you have generated any custom code during the course of this investigation, please make it available without restrictions. Please ensure that the code is sufficiently well documented and reusable, and that your Data Statement in the Editorial Manager submission system accurately describes where your code can be found. 

We expect to receive your revised manuscript within 1 month, but please let us know if you need an extension given the upcoming holiday period. 

*Published Peer Review History*

*Press*

Kind regards,

Richard

Richard Hodge, PhD

rhodge@plos.org

Reviewer remarks:

Reviewer #1: The authors have made commendable efforts to address the concerns raised, which is greatly appreciated. They have resolved the majority of my queries and incorporated new, valuable data in response to other reviewers, thereby enriching the scientific value of the manuscript. As is often the case, some experiments have uncovered new information that may serve as material for future investigations. In summary, while the mechanistic details remain partially elucidated and not fully conclusive, I agree with the authors that there is sufficient evidence to support their model and interpretation of the results. This makes the manuscript suitable for publication in its current form. Its publication is particularly relevant as it provides an alternative explanation for some unconventional caspase functions, highlighting distinct activation mechanisms of Caspase-3 and -7.

Reviewer #2 (Lisa Bouchier-Hayes, signs review): The authors added new experiments to address my comments and suggestions. I view this paper as a very interesting addition to the growing body of evidence that caspases have non-apoptotic functions. What sets this article apart is that it provides mechanistic insight by identifying non canonical processing of caspase-7.

Minor comment: please check that the figure labels for Figure 2 are correctly cited in the text - I think some of the panels were mislabeled. 

Reviewer #3: The author's have addressed my concerns.

---

## [Editor Report · Decision Letter 3]

24 Jan 2025

Dear Sharon,

On behalf of my colleagues and the Academic Editor, Mathieu Bertrand, I am pleased to say that we can accept your manuscript for publication, provided you address any remaining formatting and reporting issues. These will be detailed in an email you should receive within 2-3 business days from our colleagues in the journal operations team; no action is required from you until then. Please note that we will not be able to formally accept your manuscript and schedule it for publication until you have completed any requested changes.

PRESS

Best wishes, 

Richard

Richard Hodge, PhD

rhodge@plos.org

PLOS
